# HELM: Hyperbolic Large Language Models via Mixture-of-Curvature Experts

**Neil He**[*]    **Rishabh Anand**[*]    **Hiren Madhu**    **Ali Maatouk**

**Smita Krishnaswamy**    **Leandros Tassiulas**    **Menglin Yang**[†]    **Rex Ying**

Yale University, USA    [*]Equal Contribution

Open-source code: github.com/Graph-and-Geometric-Learning/helm

## Abstract

Frontier large language models (LLMs) have shown great success in text modeling and generation tasks across domains. However, natural language exhibits inherent semantic hierarchies and nuanced geometric structure, which current LLMs do not capture completely owing to their reliance on Euclidean operations such as dot-products and norms. Furthermore, recent studies have shown that not respecting the underlying geometry of token embeddings leads to training instabilities and degradation of generative capabilities. These findings suggest that shifting to non-Euclidean geometries can better align language models with the underlying geometry of text. We thus propose to operate fully in *Hyperbolic space*, known for its expansive, scale-free, and low-distortion properties. To this end, we introduce HELM, a family of **H**yp**E**rbolic **L**arge Language **M**odels, offering a geometric rethinking of the Transformer-based LLM that addresses the representational inflexibility, missing set of necessary operations, and poor scalability of existing hyperbolic LMs. We additionally introduce a **Mi**xture-of-**C**urvature **E**xperts model, HELM-MiCE, where each expert operates in a distinct curvature space to encode more fine-grained geometric structure from text, as well as a dense model, HELM-D. For HELM-MiCE, we further develop hyperbolic Multi-Head Latent Attention (HMLA) for efficient, reduced-KV-cache training and inference. For both models, we further develop essential hyperbolic equivalents of rotary positional encodings and root mean square normalization. We are the first to train fully hyperbolic LLMs at billion-parameter scale, and evaluate them on well-known benchmarks such as MMLU and ARC, spanning STEM problem-solving, general knowledge, and commonsense reasoning. Our results show consistent gains from our HELM architectures – up to 4% – over popular Euclidean architectures used in LLaMA and DeepSeek with superior semantic hierarchy modeling capabilities, highlighting the efficacy and enhanced reasoning afforded by hyperbolic geometry in large-scale language model pretraining.

## 1   Introduction

Contemporary Large Language Models (LLMs) [18, 42, 9, 1] fundamentally operate within Euclidean space. This manifests in their reliance on Euclidean operations such as dot products and norms applied to token embeddings. However, this architecture presents a potential mismatch with the intrinsic structure of natural language data. Existing works have shown that textual data, particularly token inputs to LLMs, exhibit an inherent semantic hierarchy [48, 47, 21, 35], thus requiring a

---

[†]Correspondence Author; work done while at Yale University, USA.

39th Conference on Neural Information Processing Systems (NeurIPS 2025).

space that can naturally accommodate these relationships. An ideal LLM architecture would possess geometric alignment with the underlying structure of the data it aims to represent.

To further illustrate the unique geometry of text data, we show in Figure 1 the distribution of *Ricci Curvature* of token embeddings from popular decoder-only LLMs on 1000 diverse samples from RedPajama [46][2]. We observe that the vast majority of tokens exhibit a wide range of negative curvatures. This has also been observed by Robinson et al. [38], who investigate the subspace of token embeddings and its inherent non-Euclidean structure. As Ricci curvature measures the local geometry of a manifold, these empirical observations suggest hierarchical token structures, while the variation in curvature values suggest complex token geometry that cannot be captured by single curvature approaches. Robinson et al. [38] also show that not respecting the geometry of tokens will harm a Transformer-based LLM's generative capabilities while introducing undue training instabilities. He et al. [21]

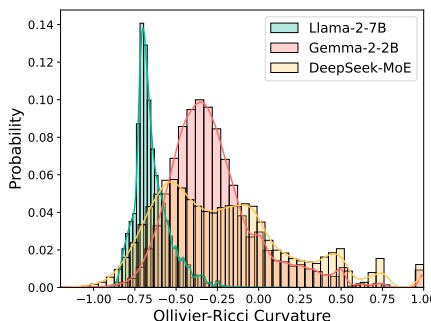

Figure 1: Ricci curvature distribution of token embeddings from decoder-only LLMs showing substantial variation of negative curvature, implying higher local hyperbolicity.

also document power law distributions of token frequencies, implying hierarchy among tokens. We thus propose a geometric rethinking of the transformer-based LLM: *operate fully in hyperbolic space*, where the negative curvature results in exponentially increasing volume w.r.t. distance. Hyperbolic space is expansive, scale-free, and yields low distortion, and has shown success in numerous settings, particularly in Transformer architectures [48, 5, 19, 6, 39]. It provides token embeddings more "breathing room" in the network, better aligning with the underlying structure of the text data.

However, previous works that study hyperbolic Transformers and pre-trained models have several major shortcomings: (1) **Inflexible geometric spaces.** They assign each Transformer block a single hyperbolic manifold, embedding entire token sequences in fixed-curvature spaces. This approach does not align with the observed substantial variant in curvature values as noted above, thus limiting the expressiveness of hidden representations. (2) **Lack of essential operations.** They omit widely-used LLM components such as rotary positional encoding and RMS normalization, and lack theoretical guarantees of LLM modules in Euclidean settings; (3) **Poor scalability.** They focus on low-dimensional settings and use quadratic hyperbolic self-attention mechanisms that do not scale comparably to modern Euclidean foundation models [18, 10]. In this work, we address the limitations of both Euclidean LLMs and prior hyperbolic Transformers through the following contributions:

- To alleviate limitation (1), we introduce hyperbolic **Mi**xture-of-**C**urvature **E**xperts (MICE), where each expert operates in a distinct curvature space, enabling the model to encode fine-grained geometric structure from text. The mixed-curvature strategy employed by MICE captures the range of negative curvatures prevalent among token embeddings, mitigating previous hyperbolic Transformers' representational inflexibility.

- To resolve limitation (2), we introduce several novel hyperbolic modules to develop hyperbolic LLMs: Hyperbolic Rotary Positional Encodings (HOPE) and Hyperbolic RMSNorm, bridging the gap between hyperbolic Transformers and modern Euclidean LLMs. Additionally, we provide extensive theoretical analysis that provides similar guarantees as in the Euclidean case.

- To address limitation (3), we propose Hyperbolic Multi-head Latent Attention (HMLA) to perform efficient inference with a reduced-footprint KV cache and reduce active memory during training, thereby bridging the scalability gap between previous hyperbolic Transformers and current Euclidean LLMs.

- Finally, we introduce HELM, the first attempt at building a family of fully **H**yp**E**rbolic **L**arge **L**anguage **M**odels. We are the first to train at billion parameter scale and outperform popular Euclidean architectures on a diverse set of benchmarks.

---

[2]Ricci Curvature is a metric of how strongly or loosely connected a region of space is. The more negative the curvature, the more hyperbolic the space is. We describe the metric in Appendix A.1.

## 2 Related Work

**Hyperbolic Transformers.** Hyperbolic geometry has emerged as a powerful embedding space for representational learning, particularly in domains characterized by hierarchical and scale-free structures [16, 34, 15]. Hyperbolic Neural Networks (HNN) [15] and their extension HNN++ [39] have shown that operating in hyperbolic spaces increases representational capacity. Several works have incorporated hyperbolic geometry into Transformers to better capture semantic relationships. HAT [19], HNN++ [39], and HyboNet [5] all propose equivalent formulations of hyperbolic attention in different *models* of hyperbolic space. HypFormer [48] developed several essential modules that were lacking in previous works, showing improved performance for structured and hierarchical datasets. Although work on hyperbolic pre-trained language models exist [6], they ignore essential components required to train *large* language models, such as normalization layers, residual connections, roatry positional encodings. Furthermore, these works suffer from the limitations mentioned in Section 1. Previous work have also considered mixed-curvature Transformers [7], however, they only using different curvature values in each attention head and relying on tangent space methods that are prone to mapping errors and numerical stability [5, 48, 20, 3]. Further related works include LResNet [23] who introduces an efficient hyperbolic residual connection method that mitigated numerical instability and tangent space mappings. Some works have devised hybrid approaches by incorporating hyperbolic components to existing pre-trained Euclidean LLMs and vision models [12, 36, 47, 17, 49, 31]. However, these works do not develop and pre-train hyperbolic LLMs from scratch. Our work advances this line of research by developing and scaling hyperbolic architectures to large-scale pretraining setups, and additionally introducing novel components for efficient and expressive language modeling in hyperbolic space.

**Open Large Language Models.** The recent surge in open LLMs has democratized access to cutting-edge NLP capabilities, enabling broader research and application development. Notable among these is LLaMA [44], which introduced a family of efficient and powerful models trained on diverse, large-scale corpora. Llama models employ several optimizations such as rotary positional embeddings and grouped query attention, making them competitive across various downstream tasks. Building on these ideas, Gemma [42] introduced further improvements for better data curation, advanced pertaining techniques, and careful model scaling strategies. In parallel, Mixture-of-Experts (MoE) architectures have emerged as a prominent design to enhance model capacity without a proportional increase in computation cost. DeepSeek-MoE [11] introduces an efficient routing mechanism to dynamically activate a subset of experts per input, significantly improving inference throughput compared to other MoE models such as Mixtral [27]. However, all these models are inherently Euclidean, and while effective, may not align well with the geometry of the token subspace.

## 3 Preliminary

We introduce the relevant background required to understand the building blocks of HELM, including hyperbolic geometry, hyperbolic self-attention, and other useful hyperbolic modules.

### 3.1 Lorentz Hyperbolic Geometry

There are several isometric models of hyperbolic space employed in previous research [45, 19, 43, 34, 48, 5, 3]. In this study, we choose the *Lorentz model*: its special space-time interactions allow for *fully hyperbolic operations*, offering enhanced expressiveness and empirical stability from an optimization perspective [5, 48, 33]. Nevertheless, our methods can be easily reformulated for other models of hyperbolic geometry via isometric mappings.

**Lorentz model.** An $n$-dimensional Lorentz model, denoted as $\mathbb{L}^{K,n}$, is a Riemannian manifold $\mathcal{L}^n$ equipped with the Riemannian metric tensor $\mathfrak{g}_n^K = \text{diag}(-1, 1, \ldots, 1)$ and defined by a constant negative curvature $K < 0$. Each point $\mathbf{x} \in \mathbb{L}^{K,n}$ has a parametrized form $[x_t, \mathbf{x}_s]^T$, where $x_t \in \mathbb{R}$ is the *time-like* dimension and $\mathbf{x}_s \in \mathbb{R}^n$ is the *space-like* dimension. For points $\mathbf{x}, \mathbf{y} \in \mathbb{L}^{K,n}$, their *Lorentzian inner product* $\langle \mathbf{x}, \mathbf{y} \rangle_{\mathcal{L}}$ is given by $\langle \mathbf{x}, \mathbf{y} \rangle_{\mathcal{L}} = -x_t y_t + \mathbf{x}_s^T \mathbf{y}_s = \mathbf{x}^T \mathfrak{g}_n^K \mathbf{y}$. Hence, the *Lorentzian norm* $\|\|\mathbf{x}\|\|_{\mathcal{L}} := \sqrt{|\langle \mathbf{x}, \mathbf{x} \rangle_{\mathcal{L}}|}$. Formally, the point set $\mathcal{L}^n := \{\mathbf{x} \in \mathbb{R}^{n+1} : \langle \mathbf{x}, \mathbf{x} \rangle_{\mathcal{L}} = 1/K, x_t > 0\}$. The origin $\mathbf{o} \in \mathbb{L}^{K,n}$ is the point $[\sqrt{-1/K}, 0, \ldots, 0]^T$.

**Tangent space.** The tangent space at a point $\mathbf{x} \in \mathbb{L}^{K,n}$ is set of points orthogonal to $\mathbf{x}$, defined as $\mathcal{T}_{\mathbf{x}}\mathbb{L}^{K,n} = \{\mathbf{y} \in \mathbb{R}^{n+1} : \langle \mathbf{x}, \mathbf{y} \rangle_{\mathcal{L}} = 0\}$. Notably, the tangent space is isomorphic to Euclidean space.

## 3.2 Hyperbolic Neural Network Modules

Extensive work has been done to develop hyperbolic neural network modules [48, 5, 3, 45, 23, 15, 39, 19], which we introduce below.

**Lorentz Linear Layer.** Given curvatures $K_1, K_2$, and parameters $\mathbf{W} \in \mathbb{R}^{(n+1)\times m}$ and $\mathbf{b} \in \mathbb{R}^m$, the *Lorentzian linear transformation with curvature* [48] is the map $\mathrm{HLT} : \mathbb{L}^{K_1,n} \to \mathbb{L}^{K_2,m}$ given by,

$$\mathrm{HLT}(\mathbf{x}; \mathbf{W}, \mathbf{b}) = \sqrt{\frac{K_2}{K_1}} \cdot \left[ \sqrt{\|\mathbf{W}^\top \mathbf{x} + \mathbf{b}\|^2 - 1/K_2}, \mathbf{W}^\top \mathbf{x} + \mathbf{b} \right]. \tag{1}$$

**Lorentz Residual Connection.** Let $\mathbf{x}, f(\mathbf{x}) \in \mathbb{L}^{K,n}$ where $\mathbf{x}$ is an input vector and $f(\mathbf{x})$ is the output of a neural network $f$. Then, the *Lorentzian residual connection* [23] is given by,

$$\mathbf{x} \oplus_{\mathcal{L}} f(\mathbf{x}) = \alpha_1 \mathbf{x} + \alpha_2 \mathbf{y}, \alpha_i = w_i / \left( \sqrt{-K} \| \| w_1 \mathbf{x} + w_2 f(\mathbf{x}) \|_{\mathcal{L}} \right) \text{ for } i \in \{0,1\}, \tag{2}$$

where $\alpha_1, \alpha_2$ are weights parametrized by constants $(w_1, w_2) \in \mathbb{R}^2 \setminus \{(0,0)\}$.

**Hyperbolic self-attention.** Hyperbolic self-attention is defined equivalently in different models through *manifold midpoints* [39, 19, 5]. Given $T$ tokens $\mathbf{X}$ where $\mathbf{x}_i \in \mathbb{L}^{K,n}$, $\mathbf{W}^{\mathbf{Q}}, \mathbf{W}^{\mathbf{K}}, \mathbf{W}^{\mathbf{V}} \in \mathbb{R}^{(n+1)\times d}$, then the *Lorentzian self-attention* [5, 48] is given by

$$\mathrm{Att}_{\mathcal{L}}(\mathbf{x})_i = \frac{\sum_{j=1}^T \nu_{i,j} \mathbf{v}_j}{\sqrt{-K} \| \| \sum_{l=1}^T \nu_{i,k} \mathbf{v}_l \|_{\mathcal{L}}}, \qquad \text{where } \nu_{i,j} = \frac{\exp\left(-d_{\mathcal{L}}^2(\mathbf{q}_i, \mathbf{k}_j)/\sqrt{m}\right)}{\sum_{l=1}^T \exp\left(-d_{\mathcal{L}}^2(\mathbf{q}_i, \mathbf{k}_l)/\sqrt{m}\right)} \tag{3}$$

where $d_{\mathcal{L}}$ is the Lorentzian distance and $\mathbf{Q} = \mathrm{HLT}(\mathbf{X}; \mathbf{W}^{\mathbf{Q}}, \mathbf{b}^{\mathbf{Q}})$, $\mathbf{K} = \mathrm{HLT}(\mathbf{X}; \mathbf{W}^{\mathbf{K}}, \mathbf{b}^{\mathbf{K}})$, $\mathbf{V} = \mathrm{HLT}(\mathbf{X}; \mathbf{W}^{\mathbf{V}}, \mathbf{b}^{\mathbf{V}})$ are the queries, keys, and values respectively.

# 4 Method

In this section, we propose several novel hyperbolic modules that serve as building blocks for HELM. We then introduce the overall architecture of HELM: a Mixture-of-Curvature variant, HELM-MICE, and a dense variant, HELM-D, for comparison.

## 4.1 Hyperbolic Rotary Positional Encoding

Previous works that proposed positional encoding methods in hyperbolic space [5, 48, 22] are confined to only learning relative encodings. However, contemporary LLMs [18, 10, 4] have instead shifted towards *Rotary Positional Encodings* (RoPE) [40], a scalable alternative that incorporates aspects from both absolute and relative encoding methods, improving length generalization and downstream performance. We thus propose a novel *hyperbolic rotary positional encoding* (HOPE) to construct positional encodings in hyperbolic space, and prove the same theoretical guarantees as RoPE. Given $T$ tokens $\mathbf{X}$, where $\mathbf{x}_i \in \mathbb{L}^{K,d}$ ($d$ even), let $\mathbf{Q}, \mathbf{K}$ be the queries and keys as in Equation (3). The hyperbolic rotary positional encoding applied to the $i$-th token is,

$$\mathrm{HOPE}(\mathbf{z}_i) = \begin{bmatrix} \sqrt{\|\mathbf{R}_{i,\Theta}(\mathbf{z}_i)_s\|^2 - 1/K} \\ \mathbf{R}_{i,\Theta}(\mathbf{z}_i)_s \end{bmatrix}; \mathbf{R}_{i,\Theta} = \begin{pmatrix} \mathbf{R}_{i,\theta_1} & 0 & 0 & \dots & 0 \\ 0 & \mathbf{R}_{i,\theta_2} & 0 & \dots & 0 \\ \vdots & \vdots & \ddots & \dots & 0 \\ 0 & 0 & \dots & \dots & \mathbf{R}_{i,\theta_{d/2}} \end{pmatrix}, \tag{4}$$

where $\mathbf{R}_{i,\theta_l}$ is the 2D rotation matrix parameterized by angle $i\theta_l$ and $\mathbf{z}$ can be a query $\mathbf{q}_i$ or key $\mathbf{k}_j$.

Next, we study the theoretical aspects of HOPE; all proofs can be found in Appendix A. First note that since $\mathbf{R}_{i,\Theta}$ is an Euclidean rotation matrix, it isometrically preserves the (Euclidean) norm of vectors. Given the definition of the Lorentz model (Section 3.1), an equivalent expression for Equation (4) is $\mathrm{HOPE}(\mathbf{z}_i) = \begin{pmatrix} 1 & 0 \\ 0 & \mathbf{R}_{i,\Theta} \end{pmatrix} \mathbf{z}_i$, making HOPE a valid Lorentzian rotation operation (see, e.g., [5]).

**Validity.** A defining characteristic for the Euclidean RoPE is that the inner product of the encoded keys and queries is a function of only the word embeddings and their relative position. Thus, only the relative positional information is encoded [40]. For hyperbolic attention in Equation (3), the analogous is defined with $-d_{\mathcal{L}}^2$ instead, which we formalize below.

**Proposition 4.1.** *Let* $\mathbf{X}$ *be* $T$ *tokens with* $\mathbf{x}_i \in \mathbb{L}^{K,d}$. *Let* $\mathbf{Q}, \mathbf{K}$ *be queries and keys as in Equation* (3). *Then* $-d_{\mathcal{L}}^2 \left( \text{HoPE} \left( \mathbf{q}_a \right), \text{HoPE} \left( \mathbf{k}_b \right) \right) = g(\mathbf{x}_a, \mathbf{x}_b; a - b)$ *for some function* $g$.

HoPE only encodes relative positional information based on Proposition 4.1 similar to RoPE, which establishes its *validity* as a RoPE operation.

**Long-term decay.** A desiring property of RoPE is long-term decay, where the attention score between a key-query pair decays when the relative position increases. HoPE has the same property as well, as shown by the following proposition.

**Proposition 4.2.** *Let* $\mathbf{Q}, \mathbf{K}$ *be as defined in Equation* (3), *then the negative square Lorentz distance* $-d_{\mathcal{L}} \left( \text{HoPE} \left( \mathbf{q}_i \right), \text{HoPE}(\mathbf{k}_j) \right)$ *can be upper bounded by* $f(\mathbf{q}_i, \mathbf{k}_j) g(i - j) < 0$, *where* $f$ *has no dependencies on position, and* $g$ *depends entirely on relative position and scales inversely w.r.t.* $i - j$.

Thus, HoPE ensures far-apart tokens have weaker connections based on Proposition 4.2, a property **not guaranteed** by previous learned encoding methods.

**Robustness to arbitrary token distances.** Barbero et al. [2] recently show that decaying token dependency (Proposition 4.2) may not be the primary reason for RoPE's success, but rather it enables LLMs to attend to specific relative distances. Our formulation of HoPE also ensures this property:

**Proposition 4.3.** *Let* $\mathbf{Q}, \mathbf{K}$ *be as defined in Equation* (3), *then* HoPE *can be maximal at an arbitrary distance, i.e., for any relative distance* $r \in \mathbb{Z}$, *there exists a key* $\mathbf{k}_j$ *such that the softmax value is maximum at distance* $r$.

HoPE thus provides hyperbolic transformers the flexibility to decay token dependencies while also ensuring robust attention across arbitrary relative distances.

**Positional attention.** Barbero et al. [2] also prove that using RoPE can help capture purely positional relationships via *diagonal* attention patterns, where tokens only attend to themselves, and *off-diagonal* attention patterns, where tokens attend only to their preceding neighbor. HoPE also allows for this:

**Proposition 4.4.** *Attention heads with* HoPE *can learn diagonal or off-diagonal attention patterns.*

We provide ablations comparing HoPE to prior hyperbolic positional encodings in Appendix D.

## 4.2 Hyperbolic Mixture-of-Curvature Module

Previous hyperbolic Transformer architectures fix each *block* to a single hyperbolic manifold, forcing the entire sequence to be embedded with a fixed curvature, restricting the flexibility of the hidden representations. Mixture-of-Experts (MoE) [26] used in Euclidean LLMs [10, 30, 13, 27, 9], where the model selectively activates only a subset of specialized "experts" for each token, enables LLMs to learn more diverse data patterns while remaining computationally efficient. However, the experts are still limited to one geometric space, restricting the collective granularity the experts can learn. Accommodating variable curvatures calls for a more flexible treatment: we propose the first *Mixture-of-Curvature Experts* (MICE) module, where each expert operates on a *distinct curvature space*. Let $\mathbf{x}_t \in \mathbb{L}^{K_1, n}$ be the $t$-th token input, then $\text{MiCE}_{N_r}^{N_s} : \mathbf{x}_t \in \mathbb{L}^{K,n} \to \mathbf{x}_t \in \mathbb{L}^{K,n}$, where $N_r$ is the number of routed experts and $N_s$ is the number of shared experts. First, we pass $\mathbf{x}_t$ through a gating module to obtain the gating scores for each routed expert, denoted as $g_{t,i}$ for $1 \leq N_r$, given as,

$$g_{t,i} = \frac{g'_{t,i}}{\sum_{j=1}^{N_r} g_{t,j}}; \ s_{t,j} = \text{act}((\mathbf{x}_t)_s^\top \mathbf{y}_j); \ g'_{t,j} = \begin{cases} s_{t,j}, & s_{t,j} \in \text{Topk}(\{s_{t,k}\}_{k \leq N_r}, K_r) \\ 0 & \text{otherwise} \end{cases} . \quad (5)$$

Here, $s_{t,j}$ is the token-expert affinity with activation function act, $\mathbf{y}_j$ is the centroid vector of the $i$-th routed expert, $\text{Topk}(S, A)$ picks the top $A$ values from set $S$, and $K_r$ is the number of activated experts. Then, the token is passed through each shared and routed expert. Let $\text{HFFN}_{r,i} : \mathbb{L}^{K_{r,i},m} \to \mathbb{L}^{K_{r,i},m}$ be the routed experts and $\text{HFFN}_{s,i} : \mathbb{L}^{K_{s,i},m} \to \mathbb{L}^{K_{s,i},m}$ be the shared experts, defined through hyperbolic feedforward networks. Here, the value of $K_{r,i}$ and $K_{s,i}$ can vary for each expert, i.e., **each expert lives on a distinct manifold**. To align the input's manifold and the experts' manifolds, first we project the tokens to the expert manifolds via $\mathbf{s}_{t,i} = \sqrt{K/K_{s,i}} \mathbf{x}_t$ and $\mathbf{r}_{t,i} = \sqrt{K/K_{r,i}} \mathbf{x}_t$. The projected token is passed through each expert and projected back to the input manifold, where we obtain $\mathbf{y}_{t,i} = \sqrt{K_{s,i}/K} \text{HFFN}_{r,i} (\mathbf{s}_{t,i})$ and $\mathbf{z}_{t,i} = \sqrt{K_{r,i}/K} \text{HFFN}_{r,i} (\mathbf{r}_{t,i})$.

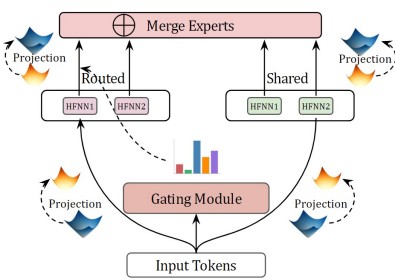

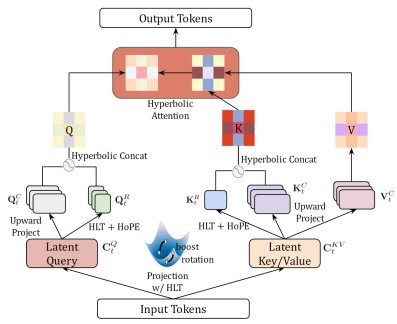

(a) **MICE module architecture.** Routed experts are selected through a gating module. The token are project from input manifold to expert manifold and then passed through each expert. The output of each expert are then project back to the input manifold and merged together through Lorentzian centroid. This modules allows experts to learn from distinct curvature spaces to allow for more granularity.

(b) **HMLA framework.** The embeddings are projected into latent space and then upward projected into queries, keys, and values. Additional decoupled queries and a shared key are created for hyperbolic positional encoding through HOPE. The queries and keys are concatenated together before performing hyperbolic self-attention.

Figure 2: Mixture-of-Curvature Experts (MICE) and hyperbolic Multi-Head Latent Attention (HMLA).

The output of $\text{MiCE}_{N_r}^{N_s}$ is given by,

$$\text{MiCE}_{N_r}^{N_s}(\mathbf{x}_t) = \mathbf{x}_t \oplus_{\mathcal{L}} \left( \frac{\sum_{i=1}^{N_s} \mathbf{y}_{t,i} + \sum_{i=1}^{N_r} \mathbf{z}_{t,i}}{\sqrt{-K} \|\| \sum_{i=1}^{N_s} \mathbf{y}_{t,i} + \sum_{i=1}^{N_r} \mathbf{z}_{t,i} \|\|_{\mathcal{L}}} \right). \tag{6}$$

The constants $\sqrt{K_{s,i}/K}, \sqrt{K_{r,i}/K}$ project from the experts' manifolds to the input manifold, ensuring that the output of each shared and routed expert lives on the same manifold. The outputs are then combined through the Lorentzian centroid [29], before performing a Lorentzian residual connection. Note that MICE is indeed a valid hyperbolic module, which we expand on in Appendix B.

### 4.3 Efficiency via Hyperbolic Multi-Head Latent Attention

Previous hyperbolic Transformers for natural language applications focus mainly on *hyperbolic self-attention*, synonymous to naive dot-product attention mechanism in Euclidean LLMs. However, recent Euclidean LLMs have gradually adopted more efficient attention methods for enhanced scalability. The quadratic attention mechanism and the extant hyperbolic self-attention formulation suffer from heavy memory challenges for large-scale training. In this section, we propose *hyperbolic Multi-Head Latent Attention (*HMLA*)* to alleviate this bottleneck. Inspired by Euclidean MLA [11, 10], HMLA reduces the size of the KV cache during inference and the active memory consumption during training. We provide a high-level description of HMLA; detailed formulation can be found in Appendix B.2.

Let $\mathbf{x}_t \in \mathbb{L}^{K,nh_n}$ be the $t$-token, where $n$ is the embedding dimension and $h_n$ is the number of heads. First, we project $\mathbf{x}_t$ via HLT to latent queries and key-value vectors, obtaining $\mathbf{c}_t^Q, \mathbf{c}_t^{KV}$ of dimensions $n_q, n_{kv}$ respectively such that $n_q, n_{kv} \ll nh_n$. We then upward-project the latent vectors back to $h_n$ heads of dimension $n$ each via HLT, obtaining $[\mathbf{k}_{t,i}^C]_{i \le h_n}, [\mathbf{v}_{t,i}^C]_{i \le h_n}$ from $\mathbf{c}_t^{KV}$, and $[\mathbf{q}_{t,i}^C]_{i \le h_n}$ from $\mathbf{c}_t^Q$. Then the projected keys are processed by positional encodings. Following previous works [11, 10], as RoPE is incompatible with MLA due to position coupling, we employ a decoupled HOPE scheme with HMLA. Through HLT, we project $\mathbf{c}_t^Q$ to additional query vectors, $[\mathbf{q}_{t,i}^R]_{i \le h_n}$, and $\mathbf{c}_t^{KV}$ to a shared key, $\mathbf{k}_t^R$, of dimension $n_r h_n$ and $n_r$ respectively ($n_r \ll nh_n$). The queries vectors, $\mathbf{q}_{t,i}^C, \mathbf{q}_{t,i}^R$, and the key vectors, $\mathbf{k}_{t,i}^C, \mathbf{k}_t^R$, are then concatenated together through Lorentzian concatenation [37, 48] for each $i$, where we obtain the final query and key vectors $\mathbf{q}_{t,i}, \mathbf{k}_{t,i}$. The attention score and output are computed using Equation (3) with $[\mathbf{q}_{t,i}]_{t \le h_n}, [\mathbf{k}_{t,i}]_{t \le h_n}, [\mathbf{v}_{t,i}^C]_{t \le h_n}$ as the queries, keys, and values, before concatenating all the heads together with Lorentzian concatenation. The concatenated result is passed through a final projection layer with HLT.

**Memory complexity.** During inference, HMLA only requires caching the latent key-value pairs. As a result, the memory footprint for HMLA is $O((n_{kv} + n_r)L)$, where $L$ is the number of layers.

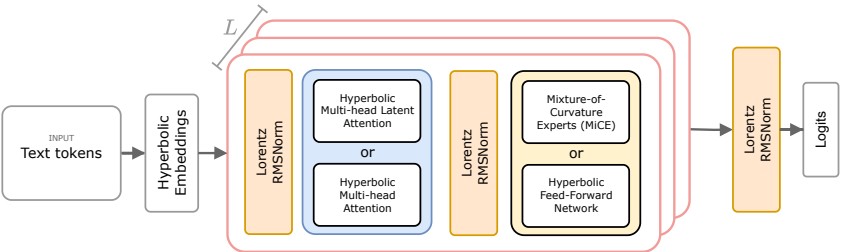

Figure 3: HELM architecture. The input tokens are mapped to hyperbolic word embeddings before being processed by a series of $L$ decoder blocks, comprising an attention block and an FFN block. The attention block (blue) can either be hyperbolic self-attention or HMLA, while the FFN block (yellow) can either be a HFFN or MɪCE layer. The output of the decoder blocks is mapped to logits. Residual connections are omitted for brevity.

In contrast, the hyperbolic self-attention used in previous hyperbolic Transformers (Equation (3)) requires storing the full-sized keys and values, resulting in a memory complexity of $O(2nn_hL)$. By choosing $n_{kv}, n_r \ll nn_h$, we have $(n_{kv}+n_r) \ll 2nn_h$, resulting in a **significantly smaller memory footprint** while maintaining the same time complexity of $O((nn_h)^2)$. Additionally, the latent query projection also results in smaller active footprint during training. This collective mechanism enables far greater scalability.

### 4.4 Hyperbolic RMSNorm

Previous works have not implemented hyperbolic root mean square normalization (RMSNorm), which is widely used in popular Euclidean LLMs [18, 11, 10] due to its stability and robustness in both forward and backward passes. Here, we propose hyperbolic RMSNorm to fill the gap:

$$\text{RMSNorm}_{\mathcal{L}}(\mathbf{x}) = \left[ \sqrt{\|\text{RMSNorm}(\mathbf{x}_s)\| - 1/K}, \text{RMSNorm}(\mathbf{x}_s) \right]^{\top}. \qquad (7)$$

Here, RMSNorm denotes the corresponding Euclidean operation. Equation (7) is equivalent to passing RMSNorm into the HRC functions from Hypformer [48]. This formulation is chosen over other methods of defining normalization layers through hyperbolic midpoint and tangent space operations [45, 3] for better numerical stability and computational efficiency.

**Invariance to input scaling.** Our formulation of hyperbolic RMSNorm is invariant to a scaling of inputs, giving us similar guarantees as Euclidean RMSNorm in terms of gradient stability during backpropagation and enhanced robustness to perturbations. We provide proofs for these guarantees in Appendix A.7.

**Proposition 4.5.** $\text{RMSNorm}_{\mathcal{L}}$ *is invariant to scaling of inputs* $\mathbf{x}$ *during both the forward and backward passes.*

### 4.5 Overall Architecture for Hyperbolic Large Language Models (HELM)

We introduce the framework for hyperbolic LLMs (HELM) based on the modules we introduced and developed in Section 4. In particular, we develop **hyperbolic LLMs with Mixture-of-Curvature Experts** (HELM-MɪCE), a class of hyperbolic LLMs with MoE modules where each expert functions in a distinct curvature space. We also construct **hyperbolic dense LLMs** (HELM-D), which resembles classic decoder-only LLMs such as Llama [18].

The overall architecture is as follows (see Figure 3): tokenized text is first mapped to learned hyperbolic word embeddings, which are then passed through a series of hyperbolic decoder blocks, each consisting of two components: 1) the attention component, where the embeddings are normalized by a $\text{RMSNorm}_{\mathcal{L}}$ layer, then processed by an attention block such as HMLA or self-attention, and finally added to the embeddings through Equation (2); and 2) the HFFN component, where the processed embeddings are again normalized by $\text{RMSNorm}_{\mathcal{L}}$ before being passed through a HFFN block and residually added to the output of the attention block (Equation (2)). For HELM-MɪCE, the

Table 1: Multichoice question-answering accuracy (%) of HELM models and their Euclidean counterparts. Bold denotes highest accuracy and underline denotes second-highest. HELM models consistently outperform their Euclidean counterparts, with HELM-MICE achieving the highest accuracy overall. The 100M model performances are average over 3 runs.

| Model | # Params | CommonsenseQA 0-Shot | HellaSwag 0-Shot | OpenbookQA 0-Shot | MMLU 5-Shot | ARC-Challenging 5-Shot | Avg - |
|---|---|---|---|---|---|---|---|
| LLaMA | 115M | $\mathbf{20.9} \pm 0.3$ | $25.1 \pm 0.3$ | $25.4 \pm 0.2$ | $23.4 \pm 0.5$ | $21.0 \pm 0.2$ | $23.2 \pm 0.2$ |
| **HELM-D** | 115M | $\underline{20.3} \pm 0.2$ | $25.9 \pm 0.1$ | $27.1 \pm 0.4$ | $\underline{25.6} \pm 0.2$ | $21.4 \pm 0.3$ | $24.1 \pm 1$ |
| DEEPSEEKV3 | 120M | $19.3 \pm 0.2$ | $25.3 \pm 0.1$ | $24.0 \pm 0.4$ | $23.9 \pm 0.3$ | $22.2 \pm 0.3$ | $22.2 \pm 0.1$ |
| **HELM-MICE** | 120M | $19.7 \pm 0.3$ | $25.9 \pm 0.2$ | $27.7 \pm 0.4$ | $24.4 \pm 0.2$ | $\underline{23.2} \pm 0.5$ | $\underline{24.1} \pm 0.1$ |
| DEEPSEEKV3 | 1B | 19.5 | $\underline{26.2}$ | $\underline{27.4}$ | 23.6 | 22.7 | 23.9 |
| **HELM-MICE** | 1B | 19.8 | **26.5** | **28.4** | **25.9** | **23.7** | **24.9** |

HFFN block can either be a dense block such as $\mathrm{HFFN}_{SG}$ or a MICE block as defined in Section 4.2, where $\mathrm{HFNN}_{SG}$ is a hyperbolic SwiGLU FFN we built to be consistent with Euclidean LLMs (see Appendix B.3 for details). HELM-D contains only dense HFFN layers. The output of the final decoder block is then normalized once again before projected to logits for next-token prediction.

# 5 Experiments

We evaluate both HELM variants' ability to answer MCQ questions in two popular benchmarks, MMLU [24] and ARC [8]. Additionally, we train an ablation HELM-MICE with constant curvature across experts, comparing with HELM-MICE models with varying curvature across experts.

**Runtime, Memory, and additional experiments.** One common concern for hyperbolic models is that they could incur significant computational overhead when compared with their Euclidean counterparts. HELM models are within 1.55X in runtime and 1.11X in memory usage of the Euclidean counterparts for both the 100M and 1B models. Additional details are shown in Appendix D. We also show additional ablation and comparison with prior hyperbolic language models, for tasks such as machine translation, in Appendix D.

## 5.1 Multichoice Benchmarking

We evaluate both HELM-MICE and HELM-D at 100M-parameter scales, across a variety of benchmarks spanning STEM problem-solving, general knowledge, and commonsense reasoning. The dense models also serve as an ablation comparison with the MICE models. We further scale the HELM-MICE to 1B parameters as the smaller HELM-MICE model outperformed HELM-D overall. Additional details regarding implementation and datasets can be found in Appendix B.

**Training Setup.** We use the LLaMA3.1-8B tokenizer [18] for all models, with a vocabulary size of 128K. For HELM-D, we use hyperbolic self-attention and $\mathrm{HFFN}_{SG}$ for the decoder block. We use 6 heads and 6 layers for the 100M model. For HELM-MICE, we use HMLA and a mixture of dense and MICE layers, each with 2 active experts and one shared expert. We use 6 heads, 6 layers, and 4 experts per layer for the 100M model, and we use 14 heads, 16 layers, and 8 experts per layer for the 1B model. The experts have curvature initiated uniformly from $-0.1$ to $-2.0$. Additionally, we incorporate the auxiliary-loss-free load balancing scheme and complementary sequence-wise auxiliary loss from DeepSeekV3 [10] to encourage load balancing among the experts. Each model was trained on a cluster of 4 NVIDIA A6000 and 4 NVIDIA A800 GPUs with model and data parallelism, where at most 4 GPUs were used by each model.

We use the English portion of the Wikipedia dataset [14] for training, comprising $\sim$6.4M rows of raw text, or roughly 5B tokens.

**Hyperbolic word embedding.** Previous works [5, 39] directly map input tokens to trained hyperbolic embeddings. However, we experienced model instability when training the 1B models with this method. Therefore, we only train the space-like dimension of the Lorentz word embeddings.

**Baselines.** We test against two popular Euclidean models: one dense model and one MoE model. For the dense model, we test HELM-D against LLaMA [18]. For the MoE model, we test HELM-MICE against DeepSeekV3 [10]. We train both baselines from scratch at the same parameter scales as their HELM counterparts, with the same dataset, tokenizer, and training setup.

Table 2: Ablation accuracy, where we compare HELM-MICE with a variant where all experts have the same curvature value, denoted as MICE-CONST. Bolding denotes the highest accuracy and underline denotes the second-highest. Euclidean DEEPSEEKV3 results are shown for reference. Overall, HELM-MICE consistently achieves the highest accuracy, while both hyperbolic models still outperform the Euclidean counterpart.

| Model | # Params | CommonsenseQA 0-Shot | HellaSwag 0-Shot | OpenbookQA 0-Shot | MMLU 5-Shot | ARC-Challenging 5-Shot | Avg - |
|---|---|---|---|---|---|---|---|
| DEEPSEEKV3 | 120M | 19.2 | 25.2 | 23.4 | 24.2 | 21.8 | 22.8 |
| MICE-CONST | 120M | **20.0** | 25.6 | 27.0 | 23.5 | 22.3 | 23.7 |
| **HELM-MICE** | 120M | 19.7 $\pm$ 0.3 | **25.9** $\pm$ 0.2 | **27.7** $\pm$ 0.4 | **24.4** $\pm$ 0.2 | **23.2** $\pm$ 0.5 | **24.1** $\pm$ 0.1 |

**Benchmarks.** We evaluate on a variety of benchmarks, including STEM and general knowledge reasoning benchmarks such as MMLU [24], ARC-Challenging [8], and OpenbookQA [32], and commonsense reasoning benchmarks such as CommonsenseQA [41] ,such HellaSwag [50]. For MMLU and ARC, we use 5-shot predictions. For CommonsenseQA, OpenbookQA, and HellaSwag, we use 0-shot prediction.

**Results.** The results are shown in Table 1. We report the accuracy of the models' abilities to answer multiple choice questions from the benchmarks. We mainly focus on comparing models within the same architectural sub-family, i.e., dense models and MoE models are separately tested against each other. Both HELM variants consistently outperform their Euclidean counterparts. In particular, *the smaller* HELM-D *model achieves higher accuracy than LLaMA on four out of the five benchmarks*, whereas the *smaller* HELM-MICE *model outperforms the smaller DeepSeekV3 model on all five benchmarks*. When comparing the ∼100M-scale HELM-D and HELM-MICE models, the latter achieves comparable performance despite using overall significantly fewer active parameters. This reflects the effectiveness of using more flexible geometry. *For the larger 1B-parameter models,* HELM-MICE *consistently outperforms the 1B DeepSeekV3 model, achieving the highest accuracy overall.* While we don't provide standard deviation from multiple runs for the 1B models as is typical of models this size, we provide results of model performance across different stages of training in Appendix D and show that HELM-MICE consistently outperform its Euclidean counterpart.

In all cases, the hyperbolic LLMs achieve better overall scores across the five benchmarks. The HELM models also always achieve higher accuracy on the *more difficult* reasoning benchmarks, namely MMLU and ARC-Challenging. This suggests better reasoning capability afforded by incorporating more suitable geometries in the embedding space. Overall, our results demonstrate the superiority of hyperbolic LLMs – in particular, the Mixture-of-Curvature Experts framework – in answering complex multiple-choice questions across a wide range of domains.

## 5.2 Ablating Distinct Curvature Learning with HELM-MICE

To assess the effectiveness of each expert operating in a distinct curvature space, we train a 120M-parameter HELM-MICE model where the curvature of each expert is fixed to $-1.0$, which we denote as MICE-CONST. Consequently, MICE-CONST embeds the entire token sequence into a fixed space of curvature $-1.0$ similar to a dense model. MICE-CONST is trained with the same setup as the preceding models. We show the results in Table 2. HELM-MICE outperforms the constant-curvature MICE-CONST in 4 out of the 5 benchmarks and achieves the higher overall accuracy, demonstrating the effectiveness of learning more expressive presentation by setting each expert to learn within a distinct manifold. Notably, MICE-CONST still outperforms the Euclidean DeepSeekV3 baseline on all 5 of the benchmarks, further demonstrating the effectiveness of hyperbolic LLMs over their Euclidean counterparts. We don't provide statistics for ablation baselines due to the need for excessive compute. Nevertheless, the results remain statistically significant as demonstrated in Table 2.

## 5.3 Qualitative Studies on Semantic Hierarchy Modeling

In this section, we qualitatively access the ability of HELM to model semantic hierarchy against its Euclidean counterparts. Our investigation of final-layer embedding distributions have found that HELM learns representations where more generic words tend to cluster in areas of smaller norm and more specific words tend to have larger norms. In Table 3, we provide case studies for HELM-MICE 1B and DeepseekV3 1B, where we show embedding norm in the final layers for words of varying

Table 3: Case study investigate of embedding norm in the final layer of HELM (1B) and DeepseekV3 (1B). Top: embedding norm of words of varying levels of specificity; Bottom: embedding norm of a question taken from the MMLU benchmark. For HELM-MiCE, more generic words are clustered closer to the origin than more specific words, which has a smaller norm than even more specific words. However, this does not necessarily hold for the DeepseekV3 1B model.

| Words | HELM-MiCE | | DeepseekV3 | |
|---|---|---|---|---|
| | Average Norm | Range | Average Norm | Range |
| to, in, have, that, and, is, for | 35.930 | 35.890~35.951 | 33.725 | 33.660~33.800 |
| study, research, subject, papers, category | 36.080 | 36.030~36.033 | 33.735 | 33.668~33.776 |
| biology, physics, chemistry, mathematics, computer science | 36.155 | 36.033~36.270 | 33.720 | 33.658~33.776 |
| algebra, geometry, photosynthesis, cellular respiration, genetics | 36.288 | 36.133~36.484 | 33.741 | 33.622~33.826 |

| HELM-MiCE | | DeepseekV3 | |
|---|---|---|---|
| Words | Norm Range | Words | Norm Range |
| A, How, does, if, there, have, is, any, with, of | 36.031~36.396 | is, a, connecting, graph, there, edges, complete, have, of | 33.668~33.768 |
| discrete, vertices, edges, connecting, pair, graph, complete, many, 10 | 36.506~36.717 | discrete, 10, how, if, pair, does, with, A, vertices, any | 33.772~33.908 |

levels of specificity (top table) and for a sample question taken from the MMLU benchmark. For HELM-MiCE, more generic words (e.g., subject) are clustered closer to the origin than more specific words (e.g., biology), which has a smaller norm than even more specific words (e.g., photosynthesis). However, this does not necessarily hold for the DeepseekV3 1B model, demonstrating how HELM-MiCE better handles semantic hierarchies. The hierarchical organization of the space enables the HELM models to sometimes better navigate the embedding space and obtain better performance.

## 6 Conclusion

In this work, we introduce HELM, a family of fully hyperbolic large language models trained at hundred-million and billion-parameter scales. Operating entirely in hyperbolic space, HELM models are better aligned with the variable geometric structure of text and token distributions. We develop MiCE modules to construct HELM-MiCE variants, enabling fine-grained geometric learning and more expressive and geometrically flexible hidden representations. We further introduce HMLA mechanism to enable HELM models to be memory efficient and improve scalability. We also introduce the HoPE and RMSNorm$_\mathcal{L}$ modules, which are fundamental to building modern hyperbolic LLMs, and support them with extensive theoretical analysis and guarantees. Trained on 5B tokens, HELM models outperform their Euclidean counterparts across benchmarks in STEM reasoning, commonsense reasoning, and general knowledge. Nevertheless, the research presented has a few limitations. Due to computational constraints, our experiments only compare HELM to Euclidean LLMs trained on the same 5B tokens, which have less representational capacity when compared to the commercially available LLMs trained on much more extensive data [4, 1, 18, 42, 10]. Additionally, we chose the Wikipedia dataset for its widely accepted reliability. However, the trained models might be under-exposed to areas such as mathematical reasoning as a result. Future work could explore incorporating scaling laws [28, 25] for hyperbolic LLMs across larger compute and data frontiers to investigate their potential.

## Acknowledgments

This work was supported in part by the National Science Foundation (NSF) IIS Div Of Information & Intelligent Systems 2403317 and Army Research Office contract W911NF-23-1-0088. We also acknowledge support in part from the Silicon Valley Community Foundation, an Amazon research award, the Yale AI Engineering Research Grant from Yale Office of the Provost, and an LEAP-U Sponsored Research from Samsung Research America. Moreover, this research has greatly benefited from the discussions and research talks held at the IMS-NTU Joint Workshop on Applied Geometry for Data Sciences. We also thank Ngoc Bui (Yale Univeristy) for useful feedback and discussion.

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

# Appendix

## A  Proofs and Details of Theoretical Results

### A.1  Ollivier-Ricci Curvature

Ricci curvature is a geometric object that measures average *geodesic dispersion* on a Riemannian manifold, i.e., whether straight paths on a surface remain parallel (zero curvature), converge (positive curvature), or diverge (negative curvature). Ollivier-Ricci curvature is the discrete analog for graphs that do not have a notion of tangent structure.

Suppose we have a graph $\mathcal{G}(V, E)$. For a node $i \in V$, we define probability measures $\mu_i$ by:

$$\mu_i(j) = \begin{cases} \frac{1}{\deg(i)} & (i, j) \in E \\ 0 & \text{otherwise.} \end{cases}$$

For $i, j \in V$, the Ollivier-Ricci curvature is given by,

$$\kappa_{\mathcal{G}}(i, j) = 1 - \frac{W_1^{\mathcal{G}}(\mu_i, \mu_j)}{d(i, j)}.$$

Here, $W_1^{\mathcal{G}}$ is the 1-Wasserstein distance between measures, $\mu_i$ and $\mu_j$. Intuitively, the curvature is defined as random walks between $i$ and $j$. If the random walks tend to stay at equal distances, $\kappa(i,j) = 0$; if they diverge, $\kappa(i,j) < 0$; and if they converge, $\kappa(i,j) > 0$. Since curvature is a local property of a Riemannian surface, for graphs, we examine local neighborhoods with a step size of 1. We thus choose $\mu_i(j) = \frac{1}{\deg(j)}$ if $(i,j) \in E$, otherwise 0.

In our preliminary analysis of popular decoder-only LLMs (Figure 1), we draw $k$-nearest-neighbors graphs with the final-layer token embeddings for a collection of prompts from RedPajama [46] to ascertain their geometric structure. We observe a high variability of negative curvature values across tokens, hinting at their non-Euclidean nature. Robinson et al. [38] further allude to the non-Euclidean token subspace, necessitating language models that can accommodate this unique geometry.

## A.2 HoPE is a Lorentz Rotation

Here, we expand on HoPE being a Lorentz rotation. Since $\mathbf{R}_{i,\Theta}$ is a Euclidean rotation, we have $\|\mathbf{z}\| = \|\mathbf{R}_{i,\Theta}\mathbf{z}\|$. Then, since for any Lorentz vector $\mathbf{z} \in \mathbb{L}^{K,n}$ we have $z_t = \sqrt{-1/K + \|\mathbf{z}_s\|} \in \mathbb{R}$. Computing $(\mathbf{R}_{i,\Theta}\mathbf{z})_t = \sqrt{-1/K + \|\mathbf{R}_{i,\Theta}\mathbf{z}_s\|} = \sqrt{-1/K + \|\mathbf{z}_s\|} = z_t$. Thus, HoPE does not affect the time-like dimension of $\mathbf{z}$, so we have,

$$\mathrm{HoPE}(\mathbf{z}) = \begin{pmatrix} 1 & 0 \\ 0 & \mathbf{R}_{i,\Theta} \end{pmatrix} \mathbf{z},$$

making HoPE a valid Lorentz rotational operation.

## A.3 Proposition 4.1: HoPE is a function of embeddings and relative position only

**Proposition.** *Let $\mathbf{X}$ be $T$ tokens with $\mathbf{x}_i \in \mathbb{L}^{K,d}$. Let $\mathbf{Q}, \mathbf{K}$ be queries and keys as in Equation (3). Then $-d_{\mathcal{L}}^2\left(\mathrm{HoPE}\left(\mathbf{q}_a\right), \mathrm{HoPE}\left(\mathbf{k}_b\right)\right)) = g(\mathbf{x}_a, \mathbf{x}_b; a - b)$ for some function $g$.*

*Proof.* We will denote $\mathrm{HoPE}(\mathbf{q_a}), \mathrm{HoPE}(\mathbf{k_b})$ as $f_q(\mathbf{x}_a), f_k(\mathbf{x}_b)$ where $f_q, f_k$ denotes the function that projects the word embeddings to queries and keys, and then applying HoPE. In practice, the projection is done through a hyperbolic linear layer, which we take to be HLT from Section 2. It suffices to prove this proposition for the case of $d = 2$, since HoPE does not affect the time-like dimension of the inputs and $\mathbf{R}_{i,\Theta}$ acts independently on each 2D block. First, note that we have

$$-d_{\mathcal{L}}^2(f_q(\mathbf{x}_a), f_k(\mathbf{x}_b)) = \frac{2}{K} - 2\langle f_q(\mathbf{x}_a), f_k(\mathbf{x}_b)\rangle_{\mathcal{L}}$$
$$= \frac{2}{K} + 2\left(f_q(\mathbf{x}_a)_t f_k(\mathbf{x}_b)_t\right) - 2\langle f_q(\mathbf{x}_a)_s, f_k(\mathbf{x}_b)_s\rangle,$$

where $\langle \cdot, \cdot \rangle_{\mathcal{L}}$ denotes Lorentzian inner product and $\langle \cdot, \cdot \rangle$ denotes the regular Euclidean inner product. Since HoPE is a Lorentz rotation, the term $2\left(f_q(\mathbf{x}_a)_t f_k(\mathbf{x}_b)_t\right)$ is simply $2((\mathbf{q}_a)_t, (\mathbf{k}_b)_t) = 2\left(\sqrt{\mathbf{W}^{\mathbf{Q}}\mathbf{x}_a - 1/K}\sqrt{\mathbf{W}^{\mathbf{K}}\mathbf{x}_b - 1/K}\right)$, so we focus on the inner product $\langle f_q(\mathbf{x}_a)_s, f_k(\mathbf{x}_b)_s \rangle$. Then, by assuming that $d = 2$, we have $(f_q(\mathbf{x}_a))_s, (f_k(\mathbf{x}_b))_s \in \mathbb{R}^2$; hence, we can parametrize these vectors by their radial and angular components. For simplicity, denote $(f_q(\mathbf{x}_a))_s, (f_k(\mathbf{x}_b))_s$ as $\mathbf{a}, \mathbf{b}$ respectively. Then, write $\langle \mathbf{a}, \mathbf{b} \rangle$ as a function $g'$. Afterwards, we parametrize the vectors as

$$\mathbf{a} = \varphi_q(\mathbf{x}_a, a)e^{i\vartheta_q(\mathbf{x}_a, a)}$$
$$\mathbf{b} = \varphi_k(\mathbf{x}_b, b)e^{i\vartheta_k(\mathbf{x}_b, b)} \tag{8}$$
$$g' = \varphi_g e^{\vartheta_{g'}},$$

where $\varphi_{\{q,k,g'\}}$ denote the radial component and $\vartheta_{\{q,k,g'\}}$ denote the angular component. Note that it suffice to show that under HoPE, we can express $\varphi_{g'}, \vartheta_{g'}$ as a function of the word embeddings and relative position. To see this, note that by definition of $g'$ we have

$$\varphi_q(\mathbf{x}_a, a)\varphi_k(\mathbf{x}_b, b) = \varphi_{g'}$$
$$\vartheta_q(\mathbf{x}_a, a) - \vartheta_k(\mathbf{x}_b, b) = \vartheta_{g'}. \tag{9}$$

Now, given the fact that HoPE acts via Euclidean rotation on the time-like dimension of any vector, we have $\varphi_q(\mathbf{x}_a, a) = \varphi_q(\mathbf{x}_a, a'), \varphi_k(\mathbf{x}_b, b) = \varphi_k(\mathbf{x}_b, b')$ for any $a', b'$. In particular, when

$a' = b' = 0$, HoPE acts via identity since all rotation angles become 0. Hence, we have

$$\varphi_q(\mathbf{x}_a, a) = \|(\mathbf{q}_a)_s\|$$
$$\varphi_k(\mathbf{x}_b, b) = \|(\mathbf{k}_b)_s\|. \tag{10}$$

Furthermore, we have $\varphi_{g'} = \|(\mathbf{q}_a)_s\|\|(\mathbf{k}_b)_s\| = \|\mathbf{W^Q x}_a\|\|\mathbf{W^K x}_b\|$ by plugging back into Equation (9), which is a function of just the word embeddings. Next, for the angular component, note that given the definition of HoPE, the rotation on any 2D block of the space-like dimension of the input at position $p$ is simply a scaling of a fixed rotation angle by $p$. Letting this fixed angle be $\sigma$, the rotation is precisely $p\sigma$. Therefore, we have

$$\vartheta_q(\mathbf{x}_a, a) = \theta_q + a\sigma$$
$$\vartheta_k(\mathbf{x}_b, b) = \theta_k + b\sigma, \tag{11}$$

where $\theta_q, \theta_k$ denote the angular components of $\mathbf{q}_a, \mathbf{k}_b$. Next, given Equation (9), we have $\vartheta_{g'} = (a - b)(\sigma) + (\theta_q - \theta_k)$. Note that we have $e^{i\theta_q} = \frac{\mathbf{W^Q x}_a}{\|\mathbf{W^Q x}_a\|}$ and $e^{i\theta_k} = \frac{\mathbf{W^K x}_b}{\|\mathbf{W^K x}_b\|}$. Consequently, $\vartheta_{g'}$ is a function of the word embeddings and the relative position $a - b$. All in all, $-d_{\mathcal{L}}^2(f_q(\mathbf{x}_a), f_k(\mathbf{x}_b))$ can be expressed with a function $g(\mathbf{x}_a, \mathbf{x}_b; a - b)$ as desired. $\square$

### A.4 Proposition 4.2: HoPE decays with increasing relative position

**Proposition.** *Let $\mathbf{Q}, \mathbf{K}$ be as defined in Equation (3), then the negative square Lorentz distance $-d_{\mathcal{L}}(\mathrm{HoPE}(\mathbf{q}_a), \mathrm{HoPE}(\mathbf{k}_b))$ can be upper bounded by $f(\mathbf{q}_a, \mathbf{k}_b)g(a - b) < 0$, where $f$ has no dependencies on position, and $g$ depends entirely on relative position and scales inversely w.r.t. $a - b$.*

*Proof.* For simplicity, we denote $\mathbf{q_a} = \mathbf{q}, \mathbf{k_b} = \mathbf{k}$. Recall that

$$-d_{\mathcal{L}}^2(\mathrm{HoPE}(\mathbf{q}), \mathrm{HoPE}(\mathbf{k})) = -\frac{2}{K} - 2(q_t k_t) + 2\mathbf{q}_s^\top \mathbf{k}_s.$$

Next, for simplicity, we denote $\mathbf{q}_s, \mathbf{k}_s$ as $\mathbf{a}, \mathbf{b}$ respectively. We group together entries of the queries and keys, where $\mathbf{a}_{[2k:2k+1]}, \mathbf{b}_{[2k:2k+1]}$ as the $2k$-th and the $(2k+1)$-th entries of $\mathbf{a}, \mathbf{b}$ respectively. With this in mind, note that since we take query and key projects to be with HLT as given in Section 2, we obtain $\mathbf{a} = \mathbf{W^Q x}_a$ and $\mathbf{b} = \mathbf{W^K x}_b$. To that end, we can assert that

$$\mathbf{a}^\top \mathbf{b} = \left(\mathbf{R}_{a,\Theta} \mathbf{W^Q x}_a\right)^\top \left(\mathbf{R}_{b,\Theta} \mathbf{W^K x}_b\right)$$
$$= \mathbf{x}_a^\top \mathbf{W^Q} \mathbf{R}_{b-a,\Theta} \mathbf{W^K x}_b$$
$$= \mathrm{Re}\left(\sum_{k=0}^{n/2} \mathbf{a}_{[2k,2k+1]} \mathbf{b}^*_{[2k,2k+1]} e^{i(b-a)\theta_k}\right), \tag{*}$$

where $\mathrm{Re}(\mathbf{x})$ denotes the real component of $\mathbf{x} \in \mathbb{C}$. Now, recall Abel's Transformation, which allows one to rewrite the sum of the product of two sequences as the product of the partial sums. Denote one of the $\mathbf{a}_{[2k,2k+1]} \mathbf{b}^*_{[2k,2k+1]}$ as $\mathbf{A}_k$, and denote the sequence $\sum_{l=0}^{k} e^{i(b-a)\theta_l}$ as $\mathbf{E}_l$. With this in mind, (*) can be written as $\mathrm{Re}\left(\sum_{k=0}^{n/2} \mathbf{A}_k(\mathbf{E}_{k+1} - \mathbf{E}_k)\right)$. Consequently, we obtain (recall that boundary term $\mathbf{A}_{n/2} = 0$)

$$|\mathbf{a}^\top \mathbf{b}| = \left|\left(\sum_{k=0}^{n/2} \mathbf{A}_k(\mathbf{E}_{k+1} - \mathbf{E}_k)\right)\right|$$
$$= \left|\left(\sum_{k=0}^{n/2} (\mathbf{A}_{k+1} - \mathbf{A}_k)\mathbf{E}_k\right)\right| \qquad \text{(Abel Transformation)}$$
$$\leq \sum_{k=0}^{n/2} |(\mathbf{A}_{k+1} - \mathbf{A}_k)||\mathbf{E}_k|$$

Now, note that the term $\mathbf{A}_{k+1} - \mathbf{A}_k$ has no dependency on position. The sum $\sum_{k=0}^{n/2} |\mathbf{E}_k|$ scales inversely with $b - a$, as shown by Su et al. [40]. To that end, we have

$$-d_{\mathcal{L}}^2 \left( \text{HoPE}\left(\mathbf{q}\right), \text{HoPE}(\mathbf{k}) \right) = -\frac{2}{K} - 2(q_t k_t) + 2\mathbf{q}_s^\top \mathbf{k}_s$$

$$\leq -\frac{2}{K} - 2(q_t k_t) + 2 \max_k |(\mathbf{A}_{k+1} - \mathbf{A}_k)| \sum_{k=0}^{n/2} |\mathbf{E}_k|.$$

Note that $\mathbf{A}_k, q_t, k_t$ depends on only the word embeddings and $\mathbf{E}_k$ depends only on the position. Thus, we have the desired result. $\qquad\square$

### A.5 Proposition 4.3: HoPE enables tokens to attend across distances

**Proposition.** *Let $\mathbf{Q}, \mathbf{K}$ be as defined in Equation (3), then HoPE can be maximal at an arbitrary distance, i.e., for any relative distance $r \in \mathbb{Z}$, there exists a key $\mathbf{k}_j$ such that the softmax value is maximum at distance $r$.*

*Proof.* We first restate Lemma A.1. from Barbero et al. [2]. The remainder of our proof follows a similar layout to that of Proposition 3.1. in Barbero et al. [2].

**Lemma A.1.** (Barbero et al. [2]) *Consider $g \in \mathbb{Q}$ with $g \neq 0$ and $n \in \mathbb{Z}$. Then, $ng \equiv 0 \pmod{2\pi}$ only when $n = 0$. In particular, this also holds if $g$ is algebraic.*

Consider a distance $r$, a non-trivial query $\mathbf{Q}_p = \psi \in \mathbb{L}^{K,n}$, as well as a key $\mathbf{K} = \text{HoPE}(\mathbf{Q}_p)$. We can represent $\psi$ as a combination of a time and space dimension on the Lorentzian manifold, $[\psi_t, \psi_s] \in \mathbb{L}^{K,n}$. Using our definition of HoPE in Section 4.1, we have

$$\mathbf{K}_q = \left[ \sqrt{\|\mathbf{R}_{p,\Theta}\psi_s\|^2 - \frac{1}{K}}, \mathbf{R}_{p,\Theta}\psi_s \right] \in \mathbb{L}^{K,n}.$$

Recall that the operator $\mathbf{R}_{p,\Theta}$ is a valid Euclidean rotation in $\mathbb{R}^n$ while HoPE remains a valid Lorentzian operation. Therefore, $\mathbf{R}_{p,\Theta}$ does not affect the Euclidean norm in the time dimension as it is an isometry. Instead, we can focus on the space dimension $\psi_s \in \mathbb{R}^n$, on which we can use the Euclidean dot product. Assume the query is at position $i$ and the key is at some $j \leq i$. We then compute the following dot product:

$$\psi_{s,p}^\top \text{HoPE}(\psi_{s,p}) = \psi_s^\top \mathbf{R}_{p,\Theta}^{(j-i)+r} \psi_s$$

$$= \sum_{l=1,\cdots,n/2} \left(\psi_s^{(l)}\right)^\top \mathbf{R}_{p,\theta_l}^{(j-i)+r} \left(\psi_s^{(l)}\right)$$

$$= \sum_{l=1,\cdots,n/2} \left\|\psi_s^{(l)}\right\|^2 \cos\left((j - i + r)\theta_l\right). \tag{12}$$

Using Lemma A.1. from Barbero et al. [2], we observe the maximum can be achieved when $j - i = -r$ for $j - i \leq 0$ since we are using causal masking, $j \leq i$. This ensures $\cos(j - i + r)\theta_l = \cos(0) = 1$, concluding the proof. $\qquad\square$

### A.6 Proposition 4.4: Attention heads with HoPE learn special positional patterns

**Proposition.** *Attention heads with HoPE can learn diagonal or off-diagonal attention patterns.*

*Proof.* The proof follows a similar layout as that of Proposition 5.3. from Barbero et al. [2]. We start with the diagonal case. Suppose $\mathbf{Q}_i = \mathbf{K}_j = \psi \in \mathbb{L}^{K,d}$, for non-trivial $\psi = [\psi_t, \psi_s]$. We assume embedding dimension $d = 2$, i.e., only a single rotation block $\mathbf{R}_{i,\theta}$ acts on the embeddings.

Recall the squared Lorentzian distance for any $a, b \in \mathbb{L}^{K,d}$,

$$
\begin{aligned}
d_{\mathcal{L}}^2(a, b) &= \|a - b\|_{\mathcal{L}}^2 \\
&= \frac{2}{K} - 2\langle a, b \rangle_{\mathcal{L}} \\
&= \frac{2}{K} - 2(-a_t b_t + \mathbf{a}_s^\top \mathbf{b}_s).
\end{aligned}
$$

Without HoPE,

$$
\begin{aligned}
-d_{\mathcal{L}}^2(\mathbf{Q}_i, \mathbf{K}_j) &= -\left[\frac{2}{K} - 2(-\psi_t \psi_t + \psi_s^\top \psi_s)\right] \\
&= -\left[\frac{2}{K} + 2\psi_t \psi_t - 2\psi_s^\top \psi_s\right].
\end{aligned}
$$

Using HoPE,

$$
\begin{aligned}
-d_{\mathcal{L}}^2(\mathbf{R}_{i,\theta}\mathbf{Q}_i, \mathbf{R}_{j,\theta}\mathbf{K}_j) &= -\left[\frac{2}{K} + 2\psi_t^2 - 2\Big((\mathbf{R}_{i,\theta}\psi_s)^\top (\mathbf{R}_{j,\theta}\psi_s)\Big)\right] \\
&= -\left[\frac{2}{K} + 2\psi_t^2 - 2\Big(\psi_s^\top \mathbf{R}_{j-i,\theta}\psi_s\Big)\right] \\
&= -\Big[\underbrace{\frac{2}{K} + 2\psi_t^2}_{C} - 2\|\psi_s\|^2 \cos\left((j-i)\theta\right)\Big] \\
&= -\Big[C - 2\|\psi_s\|^2 \cos\left((j-i)\theta\right)\Big].
\end{aligned}
$$

$$(13)$$

Using Lemma A.1. from [2], when $j = i$, we have $(j - i)\theta \equiv 0 \pmod{2\pi}$. Next, let us define $\mathbf{a}_{i,i}$ as $-[C - 2\|\psi_s\|^2]$. This means $\cos\left((j-i)\theta\right) = \cos\left(0\right) = 1$, and $\mathbf{a}_{i,j} < \mathbf{a}_{i,i}$. This gives us the following self-attention score:

$$
\begin{aligned}
\nu_{i,i} &= \frac{\exp\left(\mathbf{a}_{i,i}\right)}{\sum_{k<i} \exp\left(\mathbf{a}_{i,k}\right) + \exp\left(\mathbf{a}_{i,i}\right)} \\
&= \frac{\exp\left(-[C - 2\|\psi_s\|^2]\right)}{\sum_{k<i} \exp\left(-[C - 2\|\psi_s\|^2 \cos\left((k-i)\theta\right)]\right) + \exp\left(-[C - 2\|\psi_s\|^2]\right)} \\
&= \frac{1}{1 + \sum_{k<i} \exp\left(-[C - 2\|\psi_s\|^2 \cos\left((k-i)\theta\right)] - (-[C - r\|\psi_s^2\|^2])\right)} \\
&= \frac{1}{1 + \sum_{k<i} \exp\left(-C + 2\|\psi_s\|^2 \cos\left((k-i)\theta\right) + C - 2\|\psi_s\|^2\right)} \\
&= \frac{1}{1 + \sum_{k<i} \exp\left(2\|\psi_s\|^2(\cos\left((k-i)\theta\right) - 1)\right)},
\end{aligned}
$$

for all $k \neq i$, $\cos\left((k-i)\theta\right) < 1$. This means $2\|\psi_s\|^2(\cos\left((k-i)\theta\right) - 1) < 0$. To that end, we obtain

$$
\sup_{\|\psi_s\|^2 \to \infty} \frac{1}{1 + \sum_{k<i} \exp\left(2\|\psi_s\|^2(\cos\left((k-i)\theta\right) - 1)\right)} = 1.
$$

This guarantees $\nu_{i,i} > 1 - \epsilon$ for $\epsilon > 0$, where $\epsilon$ is a function of $2\|\psi_s\|^2$.

We now consider the off-diagonal pattern. Set $\mathbf{Q}_i = \psi$ for non-trivial $\psi = [\psi_t, \psi_s] \in \mathbb{L}^{K,d}$. Set keys $\mathbf{K}_i = \mathbf{R}_{1,\theta}\psi$ and define $\mathbf{a}_{i,i-1}$ as the off-diagonal input to the softmax when computing $\nu_{i,i-1}$. To

that end, we have

$$\mathbf{a}_{i,i-1} = -d_{\mathcal{L}}^2(\mathbf{R}_{i,\theta}\mathbf{Q}_i, \mathbf{R}_{i-1,\theta}\mathbf{K}_i)$$

$$= -\left[\frac{2}{K} + 2\psi_t^2 - 2\Big((\mathbf{R}_{i,\theta}\mathbf{Q}_i)^\top(\mathbf{R}_{i-1,\theta}\mathbf{K}_i)\Big)\right]$$

$$= -\left[\frac{2}{K} + 2\psi_t^2 - 2\Big((\mathbf{R}_{1,\theta}\psi_s)^\top(\mathbf{R}_{i-1,\theta}\mathbf{R}_{1,\theta}\psi_s)\Big)\right]$$

$$= -\left[\frac{2}{K} + 2\psi_t^2 - 2\Big((\mathbf{R}_{i,\theta}\psi_s)^\top(\mathbf{R}_{i,\theta}\psi_s)\Big)\right]$$

$$= -\left[\frac{2}{K} + 2\psi_t^2 - 2\Big(\|\psi_s\|^2\cos((i-i)\theta)\Big)\right]$$

$$= -\left[\frac{2}{K} + 2\psi_t^2 - 2\|\psi_s\|^2\right].$$

Use the same reasoning from the diagonal case to show that attention head with HoPE can learn off-diagonal patterns. This concludes the proof. □

### A.7 Proposition 4.5: Invariance guarantees of Hyperbolic RMSNorm

**Proposition.** $\mathrm{RMSNorm}_{\mathcal{L}}$ *is invariant to scaling of inputs* $\mathbf{x}$ *during both the forward and backward passes.*

*Proof.* Euclidean RMSNorm is invariant to input-scaling, both during the forward and backward pass. We observe similar guarantees from our formulation of hyperbolic RMSNorm. To that end, we first prove the input-scaling invariance of Euclidean RMSNorm.

Given an input $\mathbf{x} \in \mathbb{R}^n$ and and a feed-forward network with parameters $\mathbf{W} \in \mathbb{R}^{n \times m}$,

$$\mathbf{y} = \sigma\left(\frac{\mathbf{W}^\top\mathbf{x}}{\mathrm{RMS}(\mathbf{W}^\top\mathbf{x})} \odot \mathbf{g} + \mathbf{b}\right), \qquad \mathrm{RMS}(\mathbf{a}) = \sqrt{\frac{1}{m}\sum_{k=i}^{m}\mathbf{a}_i^2}.$$

Here, $\mathbf{g}$ is a learnable gain parameter, initially set to 1, that re-scales the standardized inputs and $\mathbf{b}$ is a bias term.

Suppose the weights are scaled by a small factor, $\mathbf{W}' = \delta\mathbf{W}$. First, observe that the root mean squared operation, RMS, is input-scaling invariant: $\mathrm{RMS}(\alpha\mathbf{a}) = \alpha\mathrm{RMS}(\mathbf{a})$. It is then evident that the final output of RMSNorm is also scale-invariant:

$$\mathbf{y}' = \sigma\left(\frac{(\mathbf{W}')^\top\mathbf{x}}{\mathrm{RMS}((\mathbf{W}')^\top\mathbf{x})} \odot \mathbf{g} + \mathbf{b}\right)$$

$$= \sigma\left(\frac{\delta\mathbf{W}^\top\mathbf{x}}{\mathrm{RMS}(\delta\mathbf{W}^\top\mathbf{x})} \odot \mathbf{g} + \mathbf{b}\right)$$

$$= \sigma\left(\frac{\cancel{\delta}\mathbf{W}^\top\mathbf{x}}{\cancel{\delta}\mathrm{RMS}(\mathbf{W}^\top\mathbf{x})} \odot \mathbf{g} + \mathbf{b}\right) = \mathbf{y} \tag{14}$$

A similar argument can be made for a scaling of the inputs $\mathbf{x}$. Since hyperbolic RMSNorm uses Euclidean RMSNorm internally, it offers the same invariance guarantees as it operates solely on the space dimension of the Lorentzian input.

Given an input $\mathbf{x} = [x_t, \mathbf{x}_s] \in \mathbb{L}^{K,n}$, we know $\mathrm{RMSNorm}(\delta\mathbf{x}) = \mathrm{RMSNorm}(\mathbf{x})$ for some scaling factor $\delta$. As such,

$$
\begin{aligned}
\mathbf{y}' &= \mathrm{RMSNorm}_{\mathcal{L}}(\delta\mathbf{x}) \\
&= \left[\sqrt{\|\mathrm{RMSNorm}(\delta\mathbf{x}_s)\| - 1/K}, \mathrm{RMSNorm}(\delta\mathbf{x}_s)\right]^{\top} \\
&= \left[\sqrt{\|\mathrm{RMSNorm}(\mathbf{x}_s)\| - 1/K}, \mathrm{RMSNorm}(\mathbf{x}_s)\right]^{\top} = \mathbf{y}.
\end{aligned}
$$

Next, we analyze the gradient stability of hyperbolic RMSNorm. In Euclidean RMSNorm, for a given loss $L$, we are interested in computing three gradients: $\frac{\partial L}{\partial \mathbf{g}}$ for the gain parameter, $\frac{\partial L}{\partial \mathbf{b}}$ for the bias, and $\frac{\partial L}{\partial \mathbf{W}}$ for the weights. We compute $\frac{\partial L}{\partial \mathbf{g}}$ and $\frac{\partial L}{\partial \mathbf{b}}$ as follows:

$$
\frac{\partial L}{\partial \mathbf{b}} = \frac{\partial L}{\partial \mathbf{v}} \cdot \frac{\partial \mathbf{v}}{\partial \mathbf{b}} \qquad\qquad \frac{\partial L}{\partial \mathbf{g}} = \frac{\partial L}{\partial \mathbf{v}} \odot \frac{\mathbf{W}^{\top}\mathbf{x}}{\mathrm{RMS}(\mathbf{W}^{\top}\mathbf{x})},
$$

where $\mathbf{v}$ denotes the inputs to the activation $\sigma$. These gradients are invariant to the scaling of Euclidean inputs $x$ and weights $\mathbf{W}$, trivially for $\frac{\partial L}{\partial \mathbf{b}}$, and due to the linearity established in Equation (14) for $\frac{\partial L}{\partial \mathbf{g}}$. Computing $\frac{\partial L}{\partial \mathbf{W}}$ is more involved due to the quadratic computation in RMS, but also provides invariance to input scaling as shown by Zhang and Sennrich [51].

Given an input $\mathbf{x} = [x_t, \mathbf{x}_s] \in \mathbb{L}^{K,n}$, we know $\frac{\partial L}{\partial \mathbf{g}}, \frac{\partial L}{\partial \mathbf{g}}$, and $\frac{\partial L}{\partial \mathbf{W}}$ are scaling-invariant in the backward pass since hyperbolic RMSNorm uses Euclidean RMSNorm. Thus, for any scaled hyperbolic input $\delta\mathbf{x} \in \mathbb{L}^{K,n}$, we get scaling invariance both in the time and space dimension during the backward pass. $\qquad\square$

## B  Additional Details

### B.1  MICE as a Lorentzian Module

In this section, we expand on the fact that MICE is indeed a valid hyperbolic module throughout. Note that since Equation (6) consists of the combination of a Lorentzian residual connection [23] and Lorentzian centroid [29], it suffices to show that the projection from input manifold to expert manifold, and the reverse projection, are valid projections between Lorentz hyperbolic spaces. In fact, it suffices to show that given $\mathbf{x} \in \mathbb{L}^{K_1,n}$, we have $\sqrt{K_1/K_2}\mathbf{x} \in \mathbb{L}^{K_2,n}$. To see this, note that

$$
\begin{aligned}
\left\langle \sqrt{K_1/K_2}\mathbf{x}, \sqrt{K_1/K_2}\mathbf{x} \right\rangle_{\mathcal{L}} &= \frac{K_1}{K_2}\langle \mathbf{x}, \mathbf{x}\rangle_{\mathcal{L}} \\
&= \frac{K_1}{K_2} \cdot \frac{1}{K_1} \qquad\qquad (\mathbf{x} \in \mathbb{L}^{K_1,n}) \\
&= \frac{1}{K_2}
\end{aligned}
$$

Thus, $\sqrt{K_1/K_2}\mathbf{x} \in \mathbb{L}^{K_2,n}$ as desired. Then, each projection via scaling by $\sqrt{K/K_{s,i}}$ and $\sqrt{K/K_{r,i}}$ indeed map the input vector $\mathbf{x}$ to the expert manifold, and the projection via $\sqrt{K_{s,i}/K}$ and $\sqrt{K_{r,i}/K}$ maps the output of the experts back to the input manifold. As a result, every vector in Equation (6) lives on the input manifold, hence the output lives in $\mathbb{L}^{K,n}$ as desired.

Additionally, note that since the squared Lorentzian distance is given by $d_{\mathcal{L}}^2(\mathbf{x}, \mathbf{y}) = 2/K - 2\langle \mathbf{x}, \mathbf{y}\rangle = 2/K + 2x_t y_t - 2\mathbf{x}_s^{\top}\mathbf{y}_s$, it scales inversely w.r.t. $\mathbf{x}_s^{\top}\mathbf{y}_s$. As a result, the gating score obtained through Equation (5) is minimizing the the squared hyperbolic distance between the input token vector $\mathbf{x}_t$ and the vector $\mathbf{y}_j$ (by viewing centroid vector $\mathbf{y}_j$ as the space-like dimension of a Lorentz hyperbolic vector). Therefore, the gating module is in fact a hyperbolic module as well.

### B.2  Hyperbolic Multi-Head Latent Attention

In this section, we provide the details for *hyperbolic Multi-Head Latent Attention (*HMLA*)*. Let $\mathbf{x}_t \in \mathbb{L}^{K,nh_n}$ be the $t$-token, where $n$ is the embedding dimension and $h_n$ is the number of heads. Let $\mathbf{W}^{DKV} \in \mathbb{R}^{(nh_n+1)\times n_{kv}}$ be the downward projection matrix of the keys and values, and

$\mathbf{W}^{DQ} \in \mathbb{R}^{(nnh+1) \times n_q}$ be the downward projection matrix of the query ($n_{kv}, n_q \ll nh_n$). We first compress the token into the latent spaces via $\mathbf{c}_t^{KV} = \text{HLT}(\mathbf{x}_t; \mathbf{W}^{\mathbf{KV}}, \mathbf{b}^{KV}) \in \mathbb{L}^{K, n_{kv}}, \mathbf{c}_t^Q = \text{HLT}(\mathbf{x}_t; \mathbf{W}^{\mathbf{Q}}, \mathbf{b}^Q) \in \mathbb{L}^{K, n_q}$. We then project the latent query, key, and value vectors back to the higher dimensional spaces. Specifically, let $\mathbf{W}^{\mathbf{UV}}, \mathbf{W}^{\mathbf{UK}} \in \mathbb{R}^{(n_{kv}+1) \times nh_n}$ be the upward projection matrix of the keys and values, and let $\mathbf{W}^{\mathbf{UQ}} \in \mathbb{R}^{(n_q+1) \times nh_n}$ be the upward projection matrix of the query. Then, the final projected keys, values, and queries are

$$[\mathbf{k}_{t,1}^C; \dots; \mathbf{k}_{t,h_n}^C] = \text{HLT}\left(\mathbf{c}_t^{KV}; \mathbf{W}^{\mathbf{UK}}, \mathbf{b}^{\mathbf{UK}}\right); [\mathbf{v}_{t,1}^C; \dots; \mathbf{v}_{t,h_n}^C] = \text{HLT}\left(\mathbf{c}_t^{KV}; \mathbf{W}^{\mathbf{UV}}, \mathbf{b}^{\mathbf{UV}}\right)$$

$$[\mathbf{q}_{t,1}^C; \dots; \mathbf{q}_{t,h_n}^C] = \text{HLT}\left(\mathbf{c}_t^Q; \mathbf{W}^{\mathbf{UQ}}, \mathbf{b}^{\mathbf{UQ}}\right). \tag{15}$$

Following previous works [11, 10], as RoPE is incompatible with MLA due to position coupling, we employ a decoupled HoPE scheme with HMLA, where we use additional query vectors with a shared key. Let $\mathbf{W}^{\mathbf{QR}} \in \mathbb{R}^{(n_q+1) \times (h_n n_r)}$ and $\mathbf{W}^{\mathbf{KR}} \in \mathbb{R}^{(n_{kv}+1) \times n_r}$ be the upward projection matrix of the decoupled queries and the shared key respectively, where $n_r$ is the dimension per head. We apply HoPE to these vectors to obtain the position-encoded vectors

$$[\mathbf{q}_{t,1}^R; \dots; \mathbf{q}_{t,h_n}^R] = \text{HoPE}\left(\text{HLT}\left(\mathbf{c}_t^Q; \mathbf{W}^{\mathbf{QR}}, \mathbf{b}^{\mathbf{QR}}\right)\right); \mathbf{k}_t^R = \text{HoPE}\left(\text{HLT}\left(\mathbf{c}_t^K; \mathbf{W}^{\mathbf{KR}}, \mathbf{b}^{\mathbf{KR}}\right)\right). \tag{16}$$

Then, we obtain the final query and key vectors as

$$\mathbf{q}_{t,i} = \text{HCat}(\mathbf{q}_{t,i}^C; \mathbf{q}_{t,i}^R); \mathbf{k}_{t,i} = \text{HCat}(\mathbf{k}_{t,i}^C; \mathbf{k}_t^R), \tag{17}$$

where HCat denotes hyperbolic concatenation [48, 37]. The attention score is computed based on negative squared Lorentz distance similar to Equation (3) as

$$\mathbf{o}_{t,i} = \frac{\sum_{j=1}^N \alpha_{t,i,j} \mathbf{v}_{t,j}^C}{\sqrt{-K} \|\|\sum_{k=1}^N \alpha_{t,i,k} \mathbf{v}_{t,j}^C\|\|_\mathcal{L}}; \alpha_{t,i,j} = \frac{\exp\left(-d_\mathcal{L}^2(\mathbf{q}_{t,i}, \mathbf{k}_{t,j})/\sqrt{h_n + n_r}\right)}{\sum_{k=1}^N \exp\left(-d_\mathcal{L}^2(\mathbf{q}_{t,i}, \mathbf{k}_{t,k})/\sqrt{h_n + n_r}\right)}. \tag{18}$$

The final output of HMLA can be expressed as the concatenation of the hyperbolic vector

$$\text{HMLA}(\mathbf{X}_t; h_n, n, n_r, n_q, n_{kv}) = \text{HLT}\left(\left[\sqrt{\|\mathbf{o}_t\| - 1/K}, \mathbf{o}_t\right]^\top; \mathbf{W}^{\mathbf{O}}, \mathbf{b}^{\mathbf{O}}\right), \tag{19}$$

where $\mathbf{o}_t = [\mathbf{o}_{t,1}, \dots, \mathbf{o}_{t,h_n}]$ and $\mathbf{W}^{\mathbf{O}} \in \mathbb{R}^{h_n(n+1) \times h_n n}$ is the out-project matrix. HMLA enables HELM models to improve computational efficiency during training and inference compared to the regular hyperbolic self-attention in Equation (3).

### B.3 Hyperbolic SwiGLU Feedforward Network

In this section, we introduce hyperbolic SwiGLU feedforward networks (FFNs), whose Euclidean formulation is widely used in LLMs [18, 10]. This differs from previous FNNs used in hyperbolic Transformers in the need for feature multiplication and activation function [5, 6, 48]. Let $\mathbf{x} \in \mathbb{L}^{K,n}$ be the input tokens, $\mathbf{W}_1, \mathbf{W}_3 \in \mathbb{R}^{(n+1) \times m}$ be the weights of internal projection layers and $\mathbf{W}_2 \in \mathbb{R}^{(m+1) \times n}$ be the weight matrix of the outward projection layer. Then, the hyperbolic SwiGLU FNN $\text{HFFN}_{SG} : \mathbb{L}^{K,n} \to \mathbb{L}^{K,n}$ is given by

$$\text{HFNN}_{SG}(\mathbf{x}) = \text{HLT}\left(\left[\sqrt{\|\mathbf{y}\| - 1/K}, \mathbf{y}\right]^\top; \mathbf{W}_2, \mathbf{b}_2\right)$$

$$\mathbf{y} = \text{SiLU}_\mathcal{L}(\text{HLT}(\mathbf{x}; \mathbf{W}_1, \mathbf{b}_1)) \otimes_s \text{HLT}(\mathbf{x}; \mathbf{W}_3, \mathbf{b}_3), \tag{20}$$

where $\text{SiLU}_\mathcal{L}$ denotes SiLU activation using the HRC activation operations from Hypformer [48] and $\otimes_s$ denotes multiplication on the space dimention of a Lorentz vector, i.e. $\mathbf{x} \otimes_s \mathbf{y} = \mathbf{x}_s \mathbf{y}_s$.

## C Training and Evaluation Details

In this section we detail the training and evaluation setup for the experiments.

## C.1 Models Setup

Here we detail the model setup for all the models we used in the experiments.

**HELM-MICE model setup.** We follow the notation in Appendix B.2 and Section 4.2. For HELM-MICE, we used HMLA as the attention mechanism in the attention block, MICE as the sparse feedforward network, and HFNN$_{SG}$ as the dense feedforward network. For both sizes, only the first decoder block uses the dense layer and the rest of the blocks using the MICE layer. The MICE layers use HFNN$_{SG}$ as well for its feedforward component. For the dense layer, we set the intermediate dimension to be $4h_n n$. For the MICE layers, we set the intermediate dimension of HFNN$_{SG}$ as $2h_n n$.

For the $\sim 100M$ sized model, we used 6 total layers, 6 heads ($n_h = 6$) each with $n = 64$, and we set $n_{kv} = 64$, $n_r = 16$. For MICE layers, we employ 4 experts with 2 active experts per token ($N_r = 4, K_r = 2$). We use one shared expert ($N_s = 1$). For the curvatures of the routed experts, we set them to be uniform from $-0.1$ to $2.0$. The curvature of the shared expert is set to be $-1$. The curvature of the entire model is set to $-1$ as well. For HMLA layers, in practice, the upward projection matrices do not need to project back to the full dimension of $h_n n$. Due to compute constraints, we instead employ a *reduction in dimensionality* during the upward projection, where $\mathbf{W^{UK}}, \mathbf{W^{UV}} \in \mathbb{R}^{(n_{kv}+1) \times h_n n/2}$, and the outward projection matrix $\mathbf{W^O}$ projects back to the full dimensionality of the input with $\mathbf{W^O} \in \mathbb{R}^{h_n(n/2+1) \times h_n n}$.

For the $\sim 1B$ sized model, we use 16 total layers, 14 heads ($n_h = 14$) each with $n = 64$, and we set $n_{kv} = 256$, $n_r = 64$. For MICE layers, we employ 4 experts with 2 active experts per token ($N_r = 8, K_r = 2$). We use one shared expert ($N_s = 1$). For the curvatures of the routed experts, we set them to be uniform from $-0.1$ to $2.0$. The curvature of the shared expert is set to be $-1$. The curvature of the entire model is set to $-1$ as well. We do not use the same reduction in dimensionality during upward projection as we did in the $\sim 100M$ case to enable for more expressive attention modules.

**HELM-D model setup.** For the HELM-D model, we only train the $100M$ sized model. Here, we use 6 layers, 6 heads each with dimension 64, and we set the intermediate dimension of the HFNN$_{SG}$ feedforward networks to be 4 times the total model dimension. We set the overall curvature of the model to $-1$. All hyperbolic models are built on top of HyperCore He et al. [22].

**Baseline models setup.** For the baseline models, we set them up to have identical dimensionality as the HELM models. In particular, for the LLaMA model we train, we use the same number of layers, heads, and dimensionality per head as the feedforward network. For the DeepSeek models we train, we use the same number of layers, heads, dimensionality per head, dimensionality for the feedforward network, dimensionality for the MoE modules, number of routed and shared experts, and the same dimensionality in the MLA layers.

**Hyperbolic work embeddings.** For the smaller HELM models, we map the input tokens directly to Lorentz hyperbolic vectors, which are then trained as hyperbolic parameters via Riemannian optimizers. The parameters are initialized via wrapped Gaussian normal distribution on the manifold. However, when training the $\sim 1B$ HELM-MICE model, we found this to cause training instability. As a result, for the larger model, we first map the tokens to the space-like dimension of Lorentz hyperbolic vectors, and then compute the time-like dimension of the vectors afterwards. We found this to stabilize model training.

## C.2 Training Details

**Dataset.** For the training dataset, we use the English portion of the Wikipedia dataset [14]. This dataset consists of $\sim 6.4M$ rows of data. We download the dataset directly from Huggingface. The raw text data is then passed through the LLaMA3.1-8B tokenizer [18], which has a vocabulary size of $\sim 128K$. We use a sequence length of 2048 for all models. Samples longer than 2048 tokens were broken up into multiple samples, with the trailing tailed dropped. The tokenized dataset consist of roughly $4.5B \sim 5B$ tokens. For training efficiency, as we measured the average number of tokens per sample is $\sim 700$ across the dataset, we used sample packing with a packing ratio of 3.0. Then packed samples shorted than 2048 tokens are then padded on the right.

Table 4: Runtime and peak memory usage per iteration comparison between HELM variants and Euclidean counterparts. HELM models are within 1.55X in runtime and 1.11X in memory usage of the Euclidean counterparts for both the 100M and 1B models.

| Model | # Params | Runtime | Memory Usage |
|---|---|---|---|
| LLaMA | 100M | 8.4s | 23.8 GB |
| HELM-D | 100M | 13.1s | 24.9 GB |
| DEEPSEEKV3 | 100M | 11.0s | 23.5 GB |
| HELM-MICE | 100M | 16.9s | 26.3 GB |
| DEEPSEEKV3 | 1B | 83.5s | 33.1 GB |
| HELM-MICE | 1B | 119.4s | 35.8 GB |

Table 5: Performance comparison between a 115M Hypformer model and HELM-D, where HELM-D consistently outperforms the baseline.

| Model | # Params | CommonsenseQA 0-Shot | HellaSwag 0-Shot | OpenbookQA 0-Shot | MMLU 5-Shot | ARC-Challenging 5-Shot | Avg - |
|---|---|---|---|---|---|---|---|
| HYPFORMER | 115M | 19.4 | 25.1 | 26.6 | 22.9 | **23.6** | 23.5 |
| **HELM-D** | 115M | **20.3** $\pm$ 0.2 | **25.9** $\pm$ 0.1 | **27.1** $\pm$ 0.4 | **25.6** $\pm$ 0.2 | 21.4 $\pm$ 0.3 | **24.1** $\pm$ 0.1 |

**Pipeline setup.** For training, we set up data-parallelism with Hugginface Accelerate. We use an effective batch size of $\sim 2M$ tokens (including padding). To ensure a fair comparison between the hyperbolic and Euclidean models, we use a learning rate of 2e-4 for all dense models and a learning rate of 4e-4 for the MoE and MICE models. A weight decay rate of 0.01 was used for all models. For the HELM-MICE models and the DeepSeek models, in order to balance the load between each expert, we utilize the auxiliary-loss-free load balancing strategy and the complementary sequence-wise auxiliary loss during training. The former punishes extreme load imbalance among the experts by dynamically updating a bias term during the gating module, while not needing an explicit auxiliary loss computation for better training efficiency. The latter punishes extreme load imbalance for any particular sequence. All training used a cosine annealing learning rate scheduler with a final target learning rate of $0.1\times$ the initial learning rate, with $3\%$ of the gradient update steps used as warmup steps.

**Runtime.** We empirically observe that HELM models take roughly 1.5 to 1.8 times the training of their Euclidean counterparts. For example, the larger $\sim 1B$ HELM-MICE model takes roughly 72 hours to train on 4 NVIDIA A800s while the similarly sized DeepSeekV3 model takes roughly 40 hours on the same machine.

### C.3 Evaluation Details

We use the Language Model Evaluation Harness library (github.com/EleutherAI/lm-evaluation-harness) for all evaluations, where the framework prompts the models with the answers choices to each question and picks the one with the highest likelihood value. For OpenbookQA, we convert the answer choices from full sentences to letter choices for all models, to make up for the relatively smaller model and training dataset sizes.

## D Ablation Studies and Additional Experiments

In this section, we perform additional experiments such as computational cost analysis, ablation studies to access the effectiveness of HOPE and HMLA, and comparisons with prior works of hyperbolic language models.

### D.1 Runtime and memory usage analysis

While HELM inevitably introduces computational overhead from operations that respect the curvature of the embedding space, our proposed methods are efficient in both runtime and memory usage. HELM models are within 1.55X in runtime and 1.11X in memory usage of the Euclidean counterparts for both the 100M and 1B models. In Table 4, we show that runtime and peak memory usage

Table 7: Ablation accuracy, where we compare HELM-MICE with a variant using hyperbolic Multi-Head self-Attention instead of HMLA, denoted as MICE-HMHA. Bolding denotes the highest accuracy and underline denotes the second-highest. Euclidean DEEPSEEKV3 results are shown for reference. Overall, HELM-MICE consistently achieves the highest accuracy, while both hyperbolic models still outperform the Euclidean counterpart.

| Model | # Params | CommonsenseQA 0-Shot | HellaSwag 0-Shot | OpenbookQA 0-Shot | MMLU 5-Shot | ARC-Challenging 5-Shot | Avg - |
|---|---|---|---|---|---|---|---|
| DEEPSEEKV3 | 120M | 19.2 | 25.2 | 23.4 | 24.2 | 21.8 | 22.8 |
| MICE-HMHA | 120M | 19.3 | 25.7 | 26.0 | 23.8 | **25.3** | 23.7 |
| textbfHELM-MICE | 120M | **19.7** ± 0.3 | **25.9** ± 0.2 | **27.7** ± 0.4 | **24.4** ± 0.2 | 23.2 ± 0.5 | **24.1** ± 0.1 |

Table 8: Ablation accuracy, where we compare HELM with a variants using learned relative positional encoding instead of HOPE, denoted as HELM-D-L and HELM-MICE-L. Bolding denotes the highest accuracy and underline denotes the second-highest. Euclidean results are shown for reference. Overall, HELM-MICE and HELM-D consistently achieves the higher accuracy, while both hyperbolic models still outperform the Euclidean counterpart.

| Model | # Params | CommonsenseQA 0-Shot | HellaSwag 0-Shot | OpenbookQA 0-Shot | MMLU 5-Shot | ARC-Challenging 5-Shot | Avg - |
|---|---|---|---|---|---|---|---|
| LLAMA | 115M | **21.1** | 25.3 | 25.3 | 23.8 | 21.0 | 23.3 |
| HELM-D-L | 115M | 19.7 | 25.5 | **28.6** | 23.0 | 21.8 | 23.7 |
| **HELM-D** | 115M | 20.3 ± 0.2 | **25.9** ± 0.1 | 27.1 ± 0.4 | **25.6** ± 0.2 | 21.4 ± 0.3 | **24.1** ± 0.1 |
| DEEPSEEKV3 | 120M | 19.2 | 25.2 | 23.4 | 24.2 | 21.8 | 22.8 |
| HELM-MICE-L | 120M | 19.0 | 25.5 | 27.0 | 23.0 | **25.7** | 24.0 |
| **HELM-MICE** | 120M | 19.7 ± 0.3 | **25.9** ± 0.2 | 27.7 ± 0.4 | 24.4 ± 0.2 | 23.2 ± 0.5 | **24.1** ± 0.1 |

comparisons between HELM and the Euclidean baselines for one training iteration (roughly 2M tokens). The results shown are for our exact experimental setup ran on 4 A100 GPUs, where we show the worst runtime and memory usage across the ranks. The results are averaged over 10 runs, and standard deviations are not shown since they are within 0.2 of the results.

## D.2 Comparison with Prior Hyperbolic Language Models

We performance additional experiments to compare the HELM architecture against prior works of hyperbolic language models and Transformers. As the majority of prior hyperbolic Transformers lacked essential components common in modern LLMs such as LayerNorm or positional encoding, we train a small version of HELM-D and compare the performance against these models in the machine translation task. The setup of our experiment is identical to that of Chen et al. [5]. Hypformer Yang et al. [48], on the other hand, does

Table 6: Machine translation BLEU score. HELM-D outperforms prior works of hyperbolic Transformers.

| Model | IWSLT'14 | WMT'14 |
|---|---|---|
| HAT Gulcehre et al. [19] | 23.7 | 21.8 |
| HNN++ Shimizu et al. [39] | 22.0 | 25.5 |
| HyboNet Chen et al. [5] | 25.9 | 26.2 |
| **HELM-D** | **26.3** | **26.5** |

possess all components. As a result, we compare the performance of HELM-D with Hypformer by training a 100M model on the same setup.

## D.3 Ablation for HMLA

Past works have found that in the Euclidean case, Multi-Head Latent Attention can achieve comparable and in some cases even superior performance compared to regular Multi-Head Attention [10]. Here we assess the effectiveness of HMLA against hyperbolic Multi-Head self-Attention. We train HELM-MICE with the same setup, where we replace the HMLA layers with a hyperbolic Multi-Head self-Attention layer as given in Equation (3). We denote the this model as MICE-HMHA. The results are shown in Table 7. HELM-MICE outperforms MICE-HMHA in 3 out of the 5 tasks, achieving the same accuracy for 1 task, with the MICE-HMHA achieving better accuracy in the last task. The results demonstrate the effectiveness of HELM-MICE while significantly reducing the memory footprint of the KV-cache. Both hyperbolic models still outperform the Euclidean model, demonstrating the effectiveness of HELM in general.

Table 9: Multi-choice answer accuracy for HELM-MICE 1B and DeepSeekV3 1B across training stages. HELM-MICE demonstrates consistent improvement over its Euclidean counter part.

| Model | Token Count | CommonsenseQA | HellaSwag | OpenbookQA | MMLU | ARC-Challenging | Avg |
|---|---|---|---|---|---|---|---|
| DeepseekV3 (1B) | 4B | 18.8 | 25.1 | 26.4 | 23.5 | 22.1 | 23.2 |
| HELM-MiCE (1B) | 4B | **19.5** | **26.0** | **27.0** | **25.6** | **22.9** | **24.3** |
| DeepseekV3 (1B) | 4.5B | 19.0 | 26.2 | 27.2 | 23.6 | 22.6 | 23.7 |
| HELM-MiCE (1B) | 4.5B | **19.7** | **26.4** | **27.6** | **25.5** | **23.1** | **24.5** |
| DeepseekV3 (1B) | 5B | 19.5 | 26.2 | 27.4 | 23.6 | 22.7 | 23.9 |
| HELM-MiCE (1B) | 5B | **19.8** | **26.5** | **28.4** | **25.9** | **23.7** | **24.9** |

## D.4 Ablation for HOPE

In this section we assess the effectiveness of HOPE against other hyperbolic positional encoding methods, namely the learned relative positional encoding from Hypformer [48]. We devise a variant of HELM-D, denoted as HELM-D-L and a variant of HELM-MICE, denoted as HELM-MICE-L, where each model uses the learned positional encoding instead of HOPE. The results are shown in Table 8. Overall both HELM-MICE and HELM-D outperform their counterparts that use learned positional encoding instead of HOPE. Interestly, however, HELM-MICE-L and HELM-D-L outperformed HELM-MICE and HELM-D respectively on the ARC-Challenging benchmark, possibly due to better alignment with reasoning prompts with non-uniform encodings. Nevertheless, the results demonstrate the effectiveness of HOPE over learned positional encodings in 4 out of the 5 tasks.

## D.5 Performance over Training Stages

While we don't provide standard deviation from multiple runs for the 1B models as is typical for model of this size, we provide the model performance of HELM-MICE and DeepSeekV3 1B in Table 9. HELM-MICE demonstrates consistent improvement over its Euclidean counter part, suggesting this improvement could sustain when the training corpus scales.

