# OpenReview forum: "HELM: Hyperbolic Large Language Models via Mixture-of-Curvature Experts"
_NeurIPS.cc/2025/Conference — NeurIPS 2025 poster_

### Official Review · Reviewer_Ar71 · 2025-06-30

**Clarity:** 4
**Significance:** 3
**Originality:** 3
**Rating:** 5
**Confidence:** 3

**Summary:**

This paper propose a new hyperbolic language model called HELM, which extends a Lorentz space Transformer architecture with modifications inspired by modern Euclidean Transformers, including hyperbolic rotary position encoding (HoPE), mixture-of-curvature (MiCE), hyperbolic multi-head latent attention (HMLA) and hyperbolic RMSNorm activations. These modifications are essential for hyperbolic Transformers to catch up on efficiency and capacities with their Euclidean counterpart. Experimental results show that HELM models with up-to 1B parameters out-perform Eucliean baselines on several multi-choice benchmarks when trained on 5B tokens.

**Questions:**

1. What are the expected benefits of hyperbolic models for frontier models? The hyperbolic models are motivated by the distortions observed in the Euclidean embedding space, but are there any indications of this distortion negatively affecting the frontier models in any tangible ways?
2. Is it possible to retain some experts in flat space in the MiCE structure? Euclidean space is already demonstrated as a solid foundation for numerous models. It would be interesting if MiCE can retain some computations in flat spaces to maximize its flexibility.

**Ethical Concerns:**

["NO or VERY MINOR ethics concerns only"]

**Final Justification:**

My main concern about HELM is their scalability in terms of both model size and training data size. Overall this work brings an interesting model architecture for foundation models and I would like to see it published in the conference.

**Limitations:**

yes

**Quality:**

3

**Strengths And Weaknesses:**

Strengths:
1. The modifications brought by HELM are essential for building hyperbolic LLMs. Compared to the barebone hyperbolic models in previous works, this paper presents a full-fledged hyperbolic model recipe that can serve as the basis to pre-train large scale, high performing hyperbolic LLMs.
2. The modifications are inspired by their Euclidean counterparts, and the adaptions to Lorentz space are intuitive and well-justified.
3. The mixture-of-curvature architecture extends the popular MoE architecture, and exploits the unique feature of hyperbolic space where different experts can resides in spaces with different curvatures.

Weakness:
1. The empirical results are limited to small scale experiments and the coverage of the benchmarks are limited.

1a. The model size studied ranges from 100m to 1B in the current experiments. I would like to see results on 7B/14B models. The motivation for hyperbolic models is the distortion of embeddings in Euclidean space, but it is unclear to me whether this distortion would manifest in meaningful way when we scale up the model size.

1b. The current pre-training dataset only contains 5B tokens from English Wikipedia. It would better to see more tokens in pre-training by using open-source datasets like C4. It is unclear whether the models have been trained to saturation and whether the advantages of hyperbolic model would still stand given more training budgets. Based on my personal experience, pre-training would not saturate for at least 100B tokens for 1B sized models.

1c. The paper only has results on multi-choice benchmarks. It would be better to also include some instruction following, math and coding benchmarks.

2. It is unclear how the HELM model compares to the Euclidean models efficiency wise in terms of both memory and computation.

---

> ### Author Rebuttal · Authors · 2025-07-31
>
> Thanks for your valuable comments! We’ll denote the reviewer's concerns with Q and our responses with R. We appreciate the reviewer finding our architectural and theoretical contributions compelling. The main concern of the reviewer focuses on the scalability of the HELM models, which we address below.
>
> **Q:** Regarding the size of the trained models and training corpora:
>
> - The model size studied ranges from 100m to 1B in the current experiments. I would like to see results on 7B/14B models…would distortion manifest in meaningful way?
> - The current pre-training dataset only contains 5B tokens from English Wikipedia….pre-training would not saturate for at least 100B tokens for 1B sized models.
>
> **R:** Thank you for the feedback and for sharing your personal experience! We agree that the models can benefit from increased parameter count and going beyond 100B tokens combining datasets like C4. Given how computationally demanding this is, however, we focused on the high-quality tokens in Wikipedia to showcase the merits of our architecture in providing a first step towards making such large-scale hyperbolic models a reality.
>
> As the first work to take a fully hyperbolic approach to LLMs and pretraining at the scale of billions of parameters and tokens, our work focuses on demonstrating the effectiveness of HELM under *computational expenses that are reasonable within the academic research context*. When going from 100M parameters to 1B, we found that the advantages of hyperbolic LLMs persist when compared to Euclidean baselines. Our architecture and implementation, such as the theoretical guarantees and hyperbolic word embedding mentioned in Sec. 5, enables the training stability needed to be the first to train hyperbolic models at this scale.
>
> We also do reasonably expect that the advantage of hyperbolic LLMs would not diminish with increasing parameter counts. Previous works have shown theoretically that Euclidean space will **always incur significant distortion when embedding tree-like data regardless of dimensionality and token count** [b], while hyperbolic space can achieve arbitrarily low distortion [a]. Thus, there is a theoretical upper bound on the performance of the Euclidean baselines at larger scales, which is not present for HELM models. We plan on releasing well-documented code, and we hope our architecture helps pave the path for 7B/14B models on the 100B of tokens scale to be trained in the future, in settings that are not limited by the computational resources of academic research.
>
> **Q:** The paper only has results on multi-choice benchmarks...better to also include some instruction following, math and coding benchmarks.
>
> **R:** Thank you for your valuable suggestion. We agree with the reviewer that additional benchmarks, such as instruction following, would be interesting. Due to limited computational resources in academic research settings, we focused on pretraining on the high-quality tokens from English Wikipedia, instead of much larger-scale data or post-training, such as instruction fine-tuning. We chose the benchmarks due to the simplicity of evaluation on multi-QA as they align with the Wikipedia tokens. Thus, they better reflects the true performance of the learned model, instead of random noise. These benchmarks are also wide used by previously accepted by NeurIPS[49,2], popular open LLMs[19, 11], and LLMs of similar size[52]. To achieve meaningful results in math or coding would require much larger computational resources that would be out of our compute abilities. For this reason, we focused on benchmarks that models can, in fact, answer based on the high-quality Wikipedia tokens it has been trained on.
>
> **Q:** It is unclear how the HELM model compares to the Euclidean models efficiency wise in terms of both memory and computation.
>
> **R:**  Our proposed methods are efficient in both runtime and memory usage. HELM models are within 1.55X in runtime and 1.11X in memory usage of the Euclidean counterparts for both the 100M and 1B models. In the table below, we show that runtime and memory comparisons between HELM and the Euclidean baselines for one training iteration (roughly 2M tokens). The results shown are for our exact experimental setup ran on 4 A100 GPUs, where we show the worst runtime and memory usage across the ranks. The results are averaged over 10 runs, and standard deviations are not shown since they are within 0.2 of the results. We will include the results in the revision.
>
> | Model | Runtime  | Memory Usage |
> | --- | --- | --- |
> | LLaMA 100M | 8.4s | 23.8Gb |
> | HELM-D 100M | 13.1s | 24.9GB |
> | DeepseekV3 100M | 11.0s | 23.5Gb |
> | HELM-MiCE 100M | 16.9s | 26.3Gb |
> | DeepseekV3 1B | 83.5s | 33.1Gb |
> | HELM-MiCE 1B | 119.4s | 35.8Gb |
>
> Additionally, there has been very little effort to develop optimized tools and libraries for hyperbolic foundation models [23]. In contrast, there is an abundance of the same for Euclidean models (e.g., LLM-Foundry, Flash Attention, DeepSpeed). Development of comparable hyperbolic libraries would further reduce the gap between HELM models and Euclidean baselines.
>
> Finally, while there is overhead for models of the **same size**, non-Euclidean models require **fewer dimensions** to embed complex structures when compared to Euclidean models [a,b]. This shows promise for non-Euclidean models to match the performance of Euclidean models with **fewer parameters** to build parameter-efficient foundation models despite the additional overhead. There is also potential to follow scaling law relationships between parameters and model performance. On the other hand, Euclidean models face heavy dimension-distortion tradeoffs [c].
>
> **Q:** What are the expected benefits of hyperbolic models for frontier models? Are there any indications of this distortion negatively affecting the frontier models in any tangible ways?
>
> **R:** The advantage of hyperbolic space is that it could better capture the hierarchical and scale-free properties in texts and language, as shown by prior works. To the best of our knowledge, there currently lacks conclusive work directly connecting numerical distortion to downstream performance in foundation models. Some previous works have explored how using manifolds that more accurately estimate the structure of the data can enhance performance in tasks related to GNNs and word embeddings (e.g., [d]).
>
> Nevertheless, our investigation into the final embedding distribution of our pre-trained HELM models suggests that hyperbolic LLMs can better represent the semantic hierarchies than the Euclidean baselines. Below are examples of how the 1B HELM-MiCE model organizes semantics as compared to the 1B DeepseekV3 model:
>
> |  | HELM-MiCE | HELM-MiCE | DeepseekV3 | DeepseekV3 |
> | --- | --- | --- | --- | --- |
> | Words | Average Norm | Range | Average Norm | Range |
> | to, in, have, that, and, is, for | 35.930 | 35.890~35.951 | 33.725 | 33.660~33.800 |
> | study, research, subject, papers, category | 36.080 | 36.030~36.033 | 33.735 | 33.668~33.776 |
> | biology, physics, chemistry, mathematics, computer science | 36.155 | 36.033~36.270 | 33.720 | 33.658~33.776 |
> | algebra, geometry, photosynthesis, cellular respiration, genetics | 36.288 | 36.133~36.484 | 33.741 | 33.622~33.826 |
>
> As we can see, for HELM-MiCE, more generic words (e.g., subject) are clustered closer to the origin than more specific words (e.g., biology), which has a smaller norm than even more specific words (e.g., photosynthesis). However, this does not necessarily hold for the DeepseekV3 1B model, demonstrating how HELM-MiCE better handles semantic hierarchies. We will provide a low-dimensional visualization of these embeddings in our revision.
>
> This difference manifests when embeddings questions HELM-MiCE correctly answer but DeepseekV3 struggles. For instance, below is a question in the MMLU benchmark categorized by its final embedding norm:
>
> | **HELM-MiCE** | **HELM-MiCE** | **DeepseekV3** | **DeepseekV3** |
> | --- | --- | --- | --- |
> | Words | Norm Range | Words | Norm Range |
> | A, How, does, if, there, have, is, any, with, of | 36.031~36.396 | is, a, connecting, graph, there, edges, complete, have, of,  | 33.668~33.768 |
> | discrete, vertices, edges, connecting, pair, graph, complete, many , 10 | 36.506~36.717 | discrete, 10, how, if, pair, does, with, A, vertices, any | 33.772~33.908 |
>
> For HELM-MiCE, general words have a smaller norm (e.g., how, if) while the more specific graph theory words are further away from the origin (e.g., connecting, graph). However, for DeepseekV3, general and specific words are sometimes intertwined in the embedding space. The hierarchical organization of the space enables the HELM models to sometimes better navigate the embedding space and obtain better performance.
>
> **Q:** Is it possible to retain some experts in flat space in the MiCE structure?
>
> **R:** Thank you for your valuable suggestions! We agree that incorporating Euclidean experts into MiCE is an exciting future direction. Our current implementation of MiCE initializes the experts with a variety of curvature values (from -0.1 to -2.0), where the lower curvature values can simulate a flatter space (like Euclidean space). To truly extend MiCE to Euclidean space, the projection coefficients in equation Sec. 4.2 can be derived from log and exp mappings. As the tangent space of hyperbolic spaces are universally Euclidean, Eq.6 can be extended to incorporate Euclidean space by lifting points to and from the tangent spaces. We will add these discussions to the revision.
>
> ***References:***
>
> [a] Sarkar, R. Low distortion delaunay embedding of trees in hyperbolic plane. In International Symposium on Graph Drawing, pp. 355–366. Springer, 2011.
>
> [b] Lee, J. R., Naor, A., and Peres, Y. Trees and markov convexity, 2007.
>
> [c] Matousek, J. Lectures on Discrete Geometry. Springer, 2002.
>
> [d] Gu, A., Sala, F., Gunel, B., and Re, C. Learning mixed-curvature representations in product spaces. ICLR, 2019.

---

> > ### Comment · Reviewer_Ar71 · 2025-08-04
> >
> > Thanks the authors for the response. I will keep my score and suggest to accept this paper. My main concern is how the method would perform when the model and data sizes scale up, but it is hard to clear without significant resources.

---

> > > ### Author Response · Authors · 2025-08-05
> > >
> > > Thank you for your reply — we appreciate the reviewer’s constructive feedback, their recognition of the quality of our work and results, and their recommendation for acceptance, especially given the constraints of academic research compute resources.

---

### Official Review · Reviewer_cdXH · 2025-07-01

**Clarity:** 3
**Significance:** 3
**Originality:** 3
**Rating:** 5
**Confidence:** 3

**Summary:**

The paper proposes shifting the geometry of LLM from Euclidean space to hyperbolic space. Therefore, the paper introduces HELM: a family of hyperbolic large language models. The paper also introduces a Mixture of Curvature experts model. In HELM-MICE, each expert operates in a distinct curvature space. The paper also develops hyperbolic multi-head latent attention.

**Questions:**

Please check weaknesses.

**Ethical Concerns:**

["NO or VERY MINOR ethics concerns only"]

**Final Justification:**

As the rebuttal has answered my questions, I increase my rating to accept.

**Limitations:**

The limitations are addresses in the paper.

**Paper Formatting Concerns:**

No formatting issues

**Quality:**

3

**Strengths And Weaknesses:**

# Strength:
The paper has a valid motivation. There is a clear explanation of the motivation in the introduction, starting from the distribution of the token embeddings. The motivation and research question are valid and novel.

The paper is the first to train a hyperbolic LLM at this scale.

The paper compares two existing models and shows the performance improvement.


# Weakness:

The paper provides numbers to compare with the existing models and shows that the performance improves. However, as the main motivation is to choose a geometry that better matches hierarchical nature, I would like to see a more detailed analysis of the models. It is still not completely clear why hyperbolic geometry results in a better performance. What has improved in the hyperbolic model? Which type of errors have decreased? In addition, adding qualitative analysis would bring benefits to better understand why the hyperbolic model is performing better.

I have a concern about the scalability of the proposed approach. However,  as this is the first step towards hyperbolic LLMs, the current scale is okay.

## Related work:
Missing related work: Hyperbolic space is beneficial in a lot of domains. To motivate the paper and cover the existing works, I would highly recommend citing the latest surveys:

@article{peng2021hyperbolic,
  title={Hyperbolic deep neural networks: A survey},
  author={Peng, Wei and Varanka, Tuomas and Mostafa, Abdelrahman and Shi, Henglin and Zhao, Guoying},
  journal={IEEE Transactions on pattern analysis and machine intelligence},
  volume={44},
  number={12},
  pages={10023--10044},
  year={2021},
  publisher={IEEE}
}

@article{mettes2024hyperbolic,
  title={Hyperbolic deep learning in computer vision: A survey},
  author={Mettes, Pascal and Ghadimi Atigh, Mina and Keller-Ressel, Martin and Gu, Jeffrey and Yeung, Serena},
  journal={International Journal of Computer Vision},
  volume={132},
  number={9},
  pages={3484--3508},
  year={2024},
  publisher={Springer}
}

Lines 57-66 talk about existing hyperbolic transformers; however, there are no citations provided. I would highly recommend adding citations, especially at the end of each limitation.

There is a research line of mixed curvature hyperbolic models. I would suggest explaining a bit and adding to the related work, and elaborating differences.

As the paper is using RoPE a lot, it would be beneficial to add that to the preliminaries as well.

## Equations:
What is y in equation 2?

Xj is not defined in Proposition 4.1.

The equation between lines 886 and 887 is not clear to me. Can the authors provide steps on how they yield that? The same holds for the equation between lines 912 and 913.

Proposition 4.2 talks about negative square Lorentz distance. Why is there no power 2 in the distance then?

I have a problem understanding equation 4 and where it comes from. Maybe the authors can elaborate by explaining RoPE and the HoPE.

## Writing:
As an overall suggestion, the writing is good; however, it needs proofreading and changing the text to make it easier to grasp.

Lines 16 and 17 suddenly introduce the dense model, HELM-D. It is confusing in the abstract.

The abstract talks about both models. What are both models?

Line 73 uses Hyperbolic RMSNorm. I would recommend using the full name first and introducing the abbreviation.

Lines 121-122 talk about several isometric models of hyperbolic space; however, it is not yet clear to the reader what the hyperbolic space is. As it is preliminary, I would recommend adding a short sentence about negative curvature and the difference from Euclidean geometry. I know that it is already added in the Lorentz model section. However, it is more beneficial to add a general sentence to introduce hyperbolic space first.

Add citations for the Lorentz model section as well.

Is Figure 2 referenced and explained in the text?


Line 286, please first explain what MCQ questions are to increase the readability of the paper.

The paper needs also qualitative analysis to explore the role of geometry on the LLMs.
Line 294: Appendix B -> Appendix C

---

> ### Author Rebuttal · Authors · 2025-07-31
>
> Thanks for your valuable comments! Thanks for the valuable edit suggestion for our writing as well, we will incorporate the edits into our revision. We also appreciate the reviewer for pointing out the additional related works, which we will incorporate into our revision as well. We’ll denote the reviewer's concerns with Q and our responses with R.
>
> **Q:** It is still not completely clear why hyperbolic geometry results in a better performance. What has improved in the hyperbolic model? Which type of errors have decreased? In addition, adding qualitative analysis would bring benefits to better understand why the hyperbolic model is performing better.
>
> **R:** Prior works of hyperbolic language and vision models have found that hyperbolic models better learn the hierarchies in semantic structures [13, 47, 35], where more generic words/concepts tend to cluster in areas of smaller norm and more specific words/concepts tend to have larger norms. Our investigation of final-layer embedding distributions have found that HELM learns representations in a similar way. Below are examples of how the 1B HELM-MiCE model organizes semantics as compared to the 1B DeepseekV3 model:
>
> |  | HELM-MiCE | HELM-MiCE | DeepseekV3 | DeepseekV3 |
> | --- | --- | --- | --- | --- |
> | Words | Average Norm | Range | Average Norm | Range |
> | to, in, have, that, and, is, for | 35.930 | 35.890~35.951 | 33.725 | 33.660~33.800 |
> | study, research, subject, papers, category | 36.080 | 36.030~36.033 | 33.735 | 33.668~33.776 |
> | biology, physics, chemistry, mathematics, computer science | 36.155 | 36.033~36.270 | 33.720 | 33.658~33.776 |
> | algebra, geometry, photosynthesis, cellular respiration, genetics | 36.288 | 36.133~36.484 | 33.741 | 33.622~33.826 |
>
> As we can see, for HELM-MiCE, more generic words (e.g., subject) are clustered closer to the origin than more specific words (e.g., biology), which has a smaller norm than even more specific words (e.g., photosynthesis). However, this does not necessarily hold for the DeepseekV3 1B model, demonstrating how HELM-MiCE better handles semantic hierarchies. We will provide low-dimensional visualization of these embeddings in our revision.
>
> This difference manifests when embeddings questions HELM-MiCE correctly answer but DeepseekV3 struggles. For instance, below is a question in the MMLU benchmark categorized by its final embedding norm:
>
> | **HELM-MiCE** | **HELM-MiCE** | **DeepseekV3** | **DeepseekV3** |
> | --- | --- | --- | --- |
> | Words | Norm Range | Words | Norm Range |
> | A, How, does, if, there, have, is, any, with, of | 36.031~36.396 | is, a, connecting, graph, there, edges, complete, have, of,  | 33.668~33.768 |
> | discrete, vertices, edges, connecting, pair, graph, complete, many , 10 | 36.506~36.717 | discrete, 10, how, if, pair, does, with, A, vertices, any | 33.772~33.908 |
>
> As we can see, for HELM-MiCE, general words have a smaller norm (e.g., how, if) while the more specific graph theory words are further away from the origin (e.g., connecting, graph). However, for DeepseekV3, general and specific words are sometimes intertwined in the embedding space. The hierarchical organization of the space enables the HELM models to sometimes better navigate the embedding space and obtain better performance.
>
> **Q:** The reviewer has concerns about the scalability of the proposed approach.
>
> **R:** We appreciate the reviewer recognizing that this is the first work to train hyperbolic LLMs and hyperbolic models at the scale of billions of parameters and tokens. To address the reviewer’s concerns, we compare the runtime and memory consumption of HELM models and the Euclidean baselines, which we will include in the revision. We found that our proposed methods are efficient and within 1.55X in runtime and 1.11X in memory usage of the Euclidean counterparts. In the table below, we show that runtime and memory comparisons between HELM and the Euclidean baselines for one training iteration (roughly 2M tokens). The results shown are for our exact experimental setup ran on 4 A100 GPUs, where we show the worst runtime and memory usage across the ranks. The results are averaged over 10 runs, and standard deviations are not shown since they are within 0.2 of the results.
>
> | Model | Runtime  | Memory Usage |
> | --- | --- | --- |
> | LLaMA 100M | 8.4s | 23.8Gb |
> | HELM-D 100M | 13.1s | 24.9GB |
> | DeepseekV3 100M | 11.0s | 23.5Gb |
> | HELM-MiCE 100M | 16.9s | 26.3Gb |
> | DeepseekV3 1B | 83.5s | 33.1Gb |
> | HELM-MiCE 1B | 119.4s | 35.8Gb |
>
> As we can see, HELM does not incur significant training overhead across the different model sizes. Additionally, while some efforts have tried to develop tools for hyperbolic foundation models [23], there currently still lacks libraries that optimize training hyperbolic foundation models, whereas there are many that are optimized for training Euclidean models (such as LLM-Foundry, Flash Attention, Deepspeed). Development of comparable hyperbolic libraries would further reduce the gap between HELM models and the Euclidean baselines.
>
> Finally, while there is overhead for models of the **same size**, non-Euclidean models require **fewer dimensions** to embed complex structures when compared to Euclidean models [a,b]. This enables the potential for non-Euclidean models to match the performance of Euclidean models with **fewer parameters** to build parameter-efficient foundation models, countering the computational overhead from non-Euclidean operations and offering the potential to continue the scaling law relationship between parameters and model performance. On the other hand, Euclidean models face heavy dimension-distortion tradeoffs [c].
>
> [a] Sarkar, R. Low distortion delaunay embedding of trees in hyperbolic plane. In International Symposium on Graph Drawing, pp. 355–366. Springer, 2011.
>
> [b] Lee, J. R., Naor, A., and Peres, Y. Trees and markov convexity, 2007.
>
> [c] Matousek, J. Lectures on Discrete Geometry. Springer, 2002.
>
> **Q:** What is y in equation 2?
>
> **R:** The y here is f(x), we will fix all of our typos in the revision.
>
> **Q:** Xj is not defined in Proposition 4.1?
>
> **R:** The i and j subscripts in the proposition index through the tokens, queries, and keys. We will make this clearer in the revision.
>
> **Q:** Confusion regarding equations on lines 886 and 887 and lines 912 and 913?
>
> **R:** The equation on these utilizes the simplified squared Lorentz distance formula from [28]
>
> **Q:** Why is there no power 2 in the distance in Proposition 4.2?
>
> **R:** We apologize for our typo; there should be an exponential in the distance
>
> **Q:** Where did equation 4 come from?
>
> **R:** In the case of the 2D Euclidean RoPE, the formulation depends on properties of Euclidean rotation operations. Let $R_{\theta_i}$ denote a 2D rotational matrix of angle $\theta_i$ One property of this matrix is that $R_{\theta_i}^TR_{\theta_j} = R_{\theta_i-\theta_j}$ for some other angle $\theta_j$. This identity allows relative position information to be encoded directly into the dot-product-attention: $(R_{\theta_i}q_i)^T(R_{\theta_j}k_j) = (q_i^T)(R_{\theta_i-\theta_j}k_j)$. The properties of rotational operations also enable other theoretical guarantees of RoPE. The higher-dimensional RoPE operations generalize this by putting the 2D matrices on the diagonal. Our proposed HoPE operation takes inspiration from RoPE and is based on Loretzian rotational operations (see [6]). This formulation enables HoPE to encode relative positional information similar to RoPE and achieve similar theoretical guarantees.

---

> > ### Comment · Reviewer_cdXH · 2025-08-04
> > **reply to the rebuttal**
> >
> > I thank the authors for providing a good rebuttal.
> > I have a question wrt to the hierarchy and norms. The norm tables add beneficial information and support the hierarchical motivation. I am not sure how the norm is calculated. Are the embeddings transformed back to the Euclidean space using a logarithmic map, and then the norm is calculated?

---

> > > ### Author Response · Authors · 2025-08-05
> > >
> > > Thank you for the follow up — we appreciate Reviewer cdXH for the constructive feedbacks and for acknowledging the quality of our rebuttal. The norm here is computed as the distance of the embedding to the origin for both the Euclidean and hyperbolic models. We compute the hyperbolic norm for HELM since hyperbolic distance is used to compute the hyperbolic attention scores. For HELM-MiCE, we computed the norm in this setup as the square root of the simplified square Lorentz distance from [28]. We hope that we have addressed all of the reviewer's concerns in our rebuttals.

---

> > > > ### Comment · Reviewer_cdXH · 2025-08-05
> > > >
> > > > Thanks for the answer. I like the provided tables, as they correctly show that while in the Euclidean space norm is similar in different words, there are different norms and some sort of hierarchy in hyperbolic space. I have a follow-up question. Is it valid to compare the Euclidean norms with the hyperbolic norm?

---

> > > > > ### Author Response · Authors · 2025-08-06
> > > > >
> > > > > We appreciate the reviewer's follow-up and their acknowledgement of how our results demonstrates the superiority of HELM models to learn semantic hierarchy. In regards to the norms, their importance is not whether the absolute values can be translated between different geometric spaces, but rather how the results indicates how well each model can distinguish between words of different levels of specificity. For the Euclidean baseline, as words of different specificity collide with each other in norm, the model potentially has more difficulty separating them apart and lead to incorrect predictions. For the HELM model, these words occupy different norm ranges and thus occupy different "ring bands" in the embedding space, which helps the model to distinguish generic words from specific ones. Additionally, the negative curvature of hyperbolic space causes the volume between successive "ring bands" to grow exponentially, which is much faster than the polynomial growth rate in Euclidean space. As a result, even if the Euclidean baseline were to demonstrate similar norm distribution (which it doesn't), the HELM model would still achieve greater spatial separation between word classes. Hopefully our responses are helpful to clear the reviewer's questions.

---

> > > > > ### Author Response · Authors · 2025-08-07
> > > > >
> > > > > We appreciate the reviewer's engagement and valuable suggestions from the discussions. We hope we have addressed the reviewer’s follow up questions in our rebuttal response.

---

### Official Review · Reviewer_2Div · 2025-07-02

**Clarity:** 2
**Significance:** 3
**Originality:** 3
**Rating:** 3
**Confidence:** 4

**Summary:**

This paper introduces a new family of Large Language Models (i.e. HELM) that can operate entirely in hyperbolic space. In particular, the authors propose architectural interventions to obtain a set of hyperbolic transformer models, including a scalable hyperbolic mixture-of-curvature expert model where each expert operates using a different curvature to better leverage the geometric structure of the space. The authors show that their architectural interventions allow training fully hyperbolic LLMs of up to a billion parameters, demonstrating competitive performance with Euclidean counterparts on different downstream tasks.

**Questions:**

- How do you ensure that the reported results and improvements are statistically significant?
- Could you provide qualitative examples of how your model improves upon Euclidean ones? In particular, how does HELM capture and leverage hierarchical information to deal with the natural language tasks you evaluate on?
- How does HELM empirically compare with previous hyperbolic transformers on different tasks? And why were previous hyperbolic approaches not considered in the experiments?

**Ethical Concerns:**

["NO or VERY MINOR ethics concerns only"]

**Final Justification:**

The authors have partially addressed some of the concerns in the original review, providing additional results expanding the breadth and scope of the evaluation.

However, the concerns regarding the marginal improvement over existing baselines remain.

**Limitations:**

No. The limitations are only briefly discussed in section 6 and should be better elicited in the experimental design through an in-depth error analysis and qualitative evaluation (please see weaknesses).

**Quality:**

2

**Strengths And Weaknesses:**

**Reason to accept:**
- The paper is generally well-written and easy to follow. The motivation for exploring hyperbolic representations for LLMs is clearly articulated, as well as its potential impact.
- The paper performs an important technical and architectural contribution, introducing novel components for building LLMs operating in hyperbolic space and introducing a new model (i.e., HELM). This is an important contribution that can support future research in the field.
- Experiments performed on different natural language tasks demonstrate that HELM is competitive with existing LLMs of the same size operating in Euclidean space.

**Reason to reject:**

While I find the architectural contribution compelling, I believe the experiments are currently insufficient for acceptance. Further experiments and analyses are required to validate the impact of their contribution. Specifically:

- The difference in performance with Euclidean models is minimal and not supported by statistical significance analyses, which, I think, given the relatively small difference in performance, is necessary.
- While the authors perform a comparison with Euclidean models of the same size, there is a missing comparison with previous Hyperbolic transformers such as HAT, HNN++, and HyboNet, HypFormer. I believe such a comparison is necessary (even considering different tasks and setups, such as relation prediction) to fully validate the quality of their contribution.
- Beyond quantitative comparison with Euclidean models, I believe the paper should include a more in-depth qualitative/error analysis to understand the real impact and limitations of the hyperbolic representation and the introduced architecture. One of the expected impacts of hyperbolic space should be an improved ability to capture hierarchical relationships. However, from the current empirical evaluation, this is not at all clear.

---

> ### Author Rebuttal · Authors · 2025-07-31
>
> Thanks for your valuable comments! We’ll denote the reviewer's concerns with Q and our responses with R.
>
> We appreciate that Reviewer 2Div finds our method novel and compelling. The concerns raised are mostly due to reservations about the significance of our experimental results. We focus our rebuttal below on solidifying our experiments and including the reviewer’s feedback in our new experimental results.
>
> **Q:** The difference in performance with Euclidean models is minimal and not supported by statistical significance analyses, which, I think, given the relatively small difference in performance, is necessary. How do you ensure that the reported results and improvements are statistically significant?
>
> **R:** We appreciate the reviewer raising this important point. To further strengthen our results, we performed additional training runs for the DeepseekV3 and HELM-MiCE at the 100M, where the statistics are computed over 3 runs as shown below:
>
> |  | CommonsenseQA   | HellaSwag  | OpenbookQA  | MMLU  | ARC-Challenging | Avg |
> | --- | --- | --- | --- | --- | --- | --- |
> | DeepseekV3 (100M) | $19.3\pm0.2$ | $25.3\pm 0.1$ | $24.0\pm0.4$ | $23.9\pm0.3$ | $22.2\pm0.3$ | $22.9\pm0.1$ |
> | HELM-MiCE (100M) | $19.7\pm0.3$ | $25.9\pm0.2$ | $27.7\pm0.4$ | $24.4\pm0.2$ | $23.2\pm0.5$ | $24.1\pm0.1$ |
>
> The results show that HELM-MiCE outperforms the Euclidean DeepseekV3 model with statistical significance. We will include these results, along with similar statistics for the 100M dense models, in the revision.
>
> As for going up to the 1B scale parameters, performing multiple training runs is rather compute-heavy, requiring thousands of GPU hours for multiple training sessions of both our model *and* the baselines. As performing multiple runs at the scale of billions of parameters becomes extremely expensive, it is typically evaluated for a single run across prior literature, even those that were previously accepted by NeurIPS[49,2], popular open LLMs[19, 11], and LLMs of similar size[52]. To showcase the significance and consistency of the results, though, we provide the results at different checkpoints to showcase how the performance gap is consistent between the two and remains beneficial to our framework:
>
> |  | Token Count | CommonsenseQA | HellaSwag | OpenbookQA | MMLU | ARC-Challenging | Avg |
> | --- | --- | --- | --- | --- | --- | --- | --- |
> | DeepseekV3 (1B) | 4B | 18.8 | 25.1 | 26.4 | 23.5 | 22.1 | 23.2 |
> | **HELM-MiCE (1B)** | **4B** | **19.5** | **26.0** | **27.0** | **25.6** | **22.9** | **24.3** |
> | DeepseekV3 (1B) | 4.5B | 19.0 | 26.2 | 27.2 | 23.6 | 22.6 | 23.7 |
> | **HELM-MiCE (1B)** | **4.5B** | **19.7** | **26.4** | **27.6** | **25.5** | **23.1** | **23.5** |
> | DeepseekV3 (1B) | 5B | 19.5    |  26.2 | 27.4 | 23.6  | 22.7 | 23.9 |
> | **HELM-MiCE (1B)** | **5B** | **19.8**    |  **26.5** | **28.4** | **25.9** | **23.7** |  **24.9** |
>
> The results demonstrate that the HELM-MiCE model consistently outperforms the Euclidean DeepseekV3 model across training stages, ranging from 4 to 5 billion tokens. We will incorporate these statistics in the revision.
>
> **Q:** While the authors perform a comparison with Euclidean models of the same size, there is a missing comparison with previous Hyperbolic transformers such as HAT, HNN++, and HyboNet, HypFormer. I believe such a comparison is necessary (even considering different tasks and setups, such as relation prediction) to fully validate the quality of their contribution. How does HELM empirically compare with previous hyperbolic transformers on different tasks? And why were previous hyperbolic approaches not considered in the experiments?
>
> **R:** We’d like to thank the reviewer for bringing attention to these prior works. Of these prior works for hyperbolic Transformers, three of which (namely HAT, HNN++, and HyboNet) each lacked hyperbolic counterparts of essential modules in modern LLMs. In comparison, all of these modules are available in HELM. We summarize the modules below:
>
> - HAT focuses on proposing hyperbolic self-attention mechanism without the rest of the hyperbolic Transformer architecture (e.g., LayerNorm, residual connection, positional encoding, linear layers), which is used in HELM-D where the operations are mathematically equivalent (as proven in HNN++)
> - HNN++ lacks positional encoding, layer normalization, and residual connection
> - HyboNet lacked layer normalization, and its positional encoding and residual connection could only be used in conjunction with its linear layers, as they are added as bias terms in the linear layers, greatly limiting the usability of these methods.
>
> Given these limitations, we have not considered these models in LLM baselining of HELM. For completeness, we compare our HELM-D model against these prior works in the machine translation tasks from HyboNet, where the choice of using the dense version of HELM comes from the fact that the prior works were all dense models. The results are shown below. The results for the baselines are taken from [6], and the original study did not show statistical information.
>
> |  | IWSLT’14 | WMT’14 |
> | --- | --- | --- |
> | HAT | 23.7 | 21.8 |
> | HNN++ | 22.0 | 25.5 |
> | HyboNet | 25.9 | 26.2 |
> | **HELM-D** | **26.3** | **26.5** |
>
> As for Hypformer, which does have all the required modules for language modeling, the differences that set HELM-D apart from Hypformer are:
>
> - Their positional encoding methods (relative for the former and our proposed HoPE for the latter).
> - Our proposed hyperbolic RMSNorm and the hyperbolic SwiGLU feedforward network (Appendix B.3.)
>
> For the former, we have provided an ablation study between the positional encoding method proposed in Hypformer and HoPE in our appendix (Table 4), where we created an ablation model HELM-D-L that replaced HoPE with Hypformer’s relative encoding method. HELM-D with HoPE consistently outperformed HELM-D-L. For the latter, we recognize that a direct comparison between HELM and Hypformer would be informative. Thus, we compare a 100M Hypformer model against HELM-D 100M below, where the choice of HELM variant is based on the fact that Hypformer is a dense model:
>
> |  | CommonsenseQA | HellaSwag | OpenbookQA | MMLU | ARC-Challenging | Avg |
> | --- | --- | --- | --- | --- | --- | --- |
> | Hypformer (100M) | 19.4 | 25.1 | 26.6 | 22.9 | 23.6 | 23.5 |
> | **HELM-D (100M)** | **20.1** | **25.9** | **27.0** | **24.7** | **23.5** | **24.0** |
>
> As shown above, the HELM-D model consistently outperforms Hypformer. Additionally, in comparison to Hypformer, the ablation model HELM-D-L still outperformed Hypformer in 4 out of the 5 tasks (Table 2), highlighting the effectiveness of the hyperbolic RMSNorm and hyperbolic SwiGLU feedforward network.
>
> **Q:** Could you provide qualitative examples of how your model improves upon Euclidean ones? In particular, how does HELM capture and leverage hierarchical information to deal with the natural language tasks you evaluate on?
>
> **R:** Prior works of hyperbolic language and vision models have found that hyperbolic models better learn the hierarchies in semantic structures [13, 47, 35], where more generic words/concepts tend to cluster in areas of smaller norm and more specific words/concepts tend to have larger norms. Our investigation of final-layer embedding distributions has found that HELM learns representations in a similar way. Below are examples of how the 1B HELM-MiCE model organizes semantics as compared to the 1B DeepseekV3 model:
>
> |  | HELM-MiCE | HELM-MiCE | DeepseekV3 | DeepseekV3 |
> | --- | --- | --- | --- | --- |
> | Words | Average Norm | Range | Average Norm | Range |
> | to, in, have, that, and, is, for | 35.930 | 35.890~35.951 | 33.725 | 33.660~33.800 |
> | study, research, subject, papers, category | 36.080 | 36.030~36.033 | 33.735 | 33.668~33.776 |
> | biology, physics, chemistry, mathematics, computer science | 36.155 | 36.033~36.270 | 33.720 | 33.658~33.776 |
> | algebra, geometry, photosynthesis, cellular respiration, genetics | 36.288 | 36.133~36.484 | 33.741 | 33.622~33.826 |
>
> As we can see, for HELM-MiCE, more generic words (e.g., subject) are clustered closer to the origin than more specific words (e.g., biology), which has a smaller norm than even more specific words (e.g., photosynthesis). However, this does not necessarily hold for the DeepseekV3 1B model, demonstrating how HELM-MiCE better handles semantic hierarchies. We will provide a low-dimensional visualization of these embeddings in our revision.
>
> This difference manifests when embedding questions HELM-MiCE correctly answers, but DeepseekV3 struggles with. For instance, below is a question in the MMLU benchmark categorized by its final embedding norm:
>
> | **HELM-MiCE** | **HELM-MiCE** | **DeepseekV3** | **DeepseekV3** |
> | --- | --- | --- | --- |
> | Words | Norm Range | Words | Norm Range |
> | A, How, does, if, there, have, is, any, with, of | 36.031~36.396 | is, a, connecting, graph, there, edges, complete, have, of,  | 33.668~33.768 |
> | discrete, vertices, edges, connecting, pair, graph, complete, many , 10 | 36.506~36.717 | discrete, 10, how, if, pair, does, with, A, vertices, any | 33.772~33.908 |
>
> As we can see, for HELM-MiCE, general words have a smaller norm (e.g., how, if) while the more specific graph theory words are further away from the origin (e.g., connecting, graph). However, for DeepseekV3, general and specific words are sometimes intertwined in the embedding space. The hierarchical organization of the space enables the HELM models to sometimes better navigate the embedding space and obtain better performance.

---

> > ### Comment · Area_Chair_Wts4 · 2025-08-05
> > **Reminder about Engaging in Discussion (Deadline Aug. 8th)**
> >
> > Dear Reviewer,
> >
> > This is a friendly reminder about engaging in the discussion.
> >
> > Best,
> >
> > AC

---

> > ### Comment · Reviewer_2Div · 2025-08-06
> > **Discussion**
> >
> > Thank you for your response and for providing additional results.
> >
> > I believe the additional results would surely improve the breadth and scope of the empirical analysis. At the same time, some of the concerns related to the marginal improvement remain, as confirmed by the additional results provided by the authors.

---

> > > ### Author Response · Authors · 2025-08-07
> > >
> > > We appreciate the reviewer's positive assessment of the added scope and breadth from the additional experimental results; we will include all the rebuttal results in the camera-ready. As regards to the improvements, we would like to emphasize that our proposed HELM models have achieved ***consistent improvements across model sizes, types, and tasks*** (14 out of 15 as compared to Euclidean baselines), as well as against ***all hyperbolic Transformers*** as suggested by the reviewer. Moreover, our rebuttal results (the first table in the response) demonstrate that our improvements are ***statistically significant***. The second table has also demonstrated consistent performance improvement of the 1B HELM model's over the Euclidean baseline throughout the training stages. Additionally, we also want to emphasize that the absolute percentages of improvements (up to 4%) are commonly viewed as ***notable, not marginal, improvements on the same benchmarks by recent papers accepted to NeurIPS*** [a,b,c,2] (see additional references below). Going beyond quantitative analysis, our qualitative analysis and case study demonstrate that the HELM models can ***better separate words of varying levels of specificity and capture semantic hierarchy*** than existing Euclidean architectures, which suggests inherent improvements as a result of our geometric choice and proposed methods. All in all, this showcases how our proposed HELM models provide both quantitative and qualitative improvements, thus highlighting the effectiveness of our proposed model architecture.
> > >
> > > [a] Zhengyan Shi, Adam X. Yang, Bin Wu, Laurence Aitchison, Emine Yilmaz, Aldo Lipani. 2024. Instruction Tuning With Loss Over Instructions. In NeurIPS
> > >
> > > [b] Chris Yuhao Liu, Yaxuan Wang, Jeffrey Flanigan, Yang Liu. 2024. Large Language Model Unlearning via Embedding-Corrupted Prompts. In NeurIPS
> > >
> > > [c] Michihiro Yasunaga, Antoine Bosselut, Hongyu Ren, Xikun Zhang, Christopher D Manning, Percy Liang, Jure Leskovec. 2022. Deep Bidirectional Language-Knowledge Graph Pretraining. In NeurIPS

---

> > > > ### Comment · Area_Chair_Wts4 · 2025-08-08
> > > >
> > > > Dear Reviewer 2Div,
> > > >
> > > > Today is the last day to engage in the discussion. Please answer the most recent response from the authors.
> > > >
> > > > Best,
> > > >
> > > > AC

---

### Official Review · Reviewer_1cTB · 2025-07-11

**Clarity:** 4
**Significance:** 3
**Originality:** 3
**Rating:** 5
**Confidence:** 3

**Summary:**

The paper introduces HELM, a family of fully hyperbolic large language models that re-implements every key Transformer component—positional encoding (HOPE), normalization (Hyperbolic RMSNorm), attention (HMLA), and feed-forward blocks—inside Lorentz space. It further proposes Mixture-of-Curvature Experts (MICE) so different experts operate at distinct curvatures, and scales these ideas to 100M – 1B parameters trained on 5B Wikipedia tokens. Across five reasoning benchmarks (MMLU, ARC-Chal., CommonsenseQA, HellaSwag, OpenBookQA) HELM-MICE beats matched Euclidean baselines and shows the largest gains on the hardest tasks.

**Questions:**

1. Are each expert’s curvature values fixed or learnable, and how sensitive are results to that choice?
2. Do you expect the hyperbolic–Euclidean gap to widen, stay flat, or shrink when trained on ≥100 B tokens? And what stability issues could emerge at that scale?
3. Could you provide low-dimensional projections (e.g., Poincaré-disk plots) of the learned embeddings to show how curvature and expert routing organize semantic structure?
4. What motivated using the Lorentz representation over alternatives like the Poincaré ball or Klein model, and did you observe concrete numerical or training-stability advantages?

**Ethical Concerns:**

["NO or VERY MINOR ethics concerns only"]

**Final Justification:**

I am satisfied with the authors' response. The efficiency aspect is addressed and the interpretability examples are convincing. I still have some doubts in the scaling behavior of this model. But I think based on compute budget and current results, the authors has shown the best they could achieve. I would recommend the acceptance of the paper.

**Limitations:**

Yes, the authors adequately addressed the limitations and potential negative societal impact of their work.

**Quality:**

3

**Strengths And Weaknesses:**

## Strengths

1. The paper tackles the mismatch between language’s hierarchical structure and Euclidean embeddings by operating wholly in hyperbolic space and letting different experts learn at different curvatures.
2. HOPE, Hyperbolic RMSNorm, and Lorentz-residual connections give HELM feature-parity with modern LLMs, while accompanying proofs ensure these modules retain the stability and relative-position guarantees of their Euclidean counterparts.
3. By combining HMLA with MICE, the authors train the first billion-parameter fully hyperbolic LLM; the models consistently top equally-sized Euclidean (dense and MoE) baselines, especially on MMLU and ARC-Challenging, demonstrating that the geometric shift yields real, if modest, gains.
4. The paper is well-written and easy to follow.

## Weaknesses

1. All models are trained on only 5B tokens, so absolute accuracy remains low and  improvements could shrink or vanish at larger data scales—making it unclear whether hyperbolic benefits persist in realistic pre-training regimes.
2. HELM introduces non-trivial engineering overhead (Lorentz ops, curvature routing, expert load-balancing), yet the paper gives no timing or memory benchmarks against standard implementations, leaving practitioners uncertain about the real compute-efficiency trade-offs.
3. The paper does not provide error analysis, case studies, or interpretive examples to show how hyperbolic geometry changes model behavior or what kinds of errors it corrects

---

> ### Author Rebuttal · Authors · 2025-07-31
>
> Thanks for your valuable comments! We’ll denote the reviewer's concerns with Q and our responses with R.
>
> **Q:** HELM introduces non-trivial engineering overhead (Lorentz ops, curvature routing, expert load-balancing), yet the paper gives no timing or memory benchmarks against standard implementations, leaving practitioners uncertain about the real compute-efficiency trade-offs.
>
> **R:** While HELM inevitably introduces computational overhead from operations that respect the curvature of the embedding space, our proposed methods are efficient in both runtime and memory usage. HELM models are within 1.55X in runtime and 1.11X in memory usage of the Euclidean counterparts for both the 100M and 1B models. In the table below, we show that runtime and memory comparisons between HELM and the Euclidean baselines for one training iteration (roughly 2M tokens). The results shown are for our exact experimental setup ran on 4 A100 GPUs, where we show the worst runtime and memory usage across the ranks. The results are averaged over 10 runs, and standard deviations are not shown since they are within 0.2 of the results. We will include the results in the revision.
>
> | Model | Runtime  | Memory Usage |
> | --- | --- | --- |
> | LLaMA 100M | 8.4s | 23.8Gb |
> | HELM-D 100M | 13.1s | 24.9GB |
> | DeepseekV3 100M | 11.0s | 23.5Gb |
> | HELM-MiCE 100M | 16.9s | 26.3Gb |
> | DeepseekV3 1B | 83.5s | 33.1Gb |
> | HELM-MiCE 1B | 119.4s | 35.8Gb |
>
> Additionally, there has been very little effort to develop optimized tools and libraries for hyperbolic foundation models [23]. In contrast, there is an abundance of the same for Euclidean models (e.g., LLM-Foundry, Flash Attention, DeepSpeed). Development of comparable hyperbolic libraries would further reduce the gap between HELM models and Euclidean baselines.
>
> Finally, while there is overhead for models of the **same size**, non-Euclidean models require **fewer dimensions** to embed complex structures when compared to Euclidean models [a,b]. This shows promise for non-Euclidean models to match the performance of Euclidean models with **fewer parameters** to build parameter-efficient foundation models despite the additional overhead. There is also potential to follow scaling law relationships between parameters and model performance. On the other hand, Euclidean models face heavy dimension-distortion tradeoffs [c].
>
> **Q:** The paper does not provide error analysis, case studies, or interpretive examples to show how hyperbolic geometry changes model behavior or what kinds of errors it corrects. Could you provide low-dimensional projections (e.g., Poincaré-disk plots) of the learned embeddings to show how curvature and expert routing organize semantic structure?
>
> **R:** Prior works of hyperbolic language and vision models have found that hyperbolic models better learn the hierarchies in semantic structures [13, 47, 35], where more generic words/concepts tend to cluster in areas of smaller norm and more specific words/concepts tend to have larger norms. Our investigation of final-layer embedding distributions has found that HELM learns representations in a similar way. Below are examples of how the 1B HELM-MiCE model organizes semantics as compared to the 1B DeepseekV3 model:
>
> |  | HELM-MiCE | HELM-MiCE | DeepseekV3 | DeepseekV3 |
> | --- | --- | --- | --- | --- |
> | Words | Average Norm | Range | Average Norm | Range |
> | to, in, have, that, and, is, for | 35.930 | 35.890~35.951 | 33.725 | 33.660~33.800 |
> | study, research, subject, papers, category | 36.080 | 36.030~36.033 | 33.735 | 33.668~33.776 |
> | biology, physics, chemistry, mathematics, computer science | 36.155 | 36.033~36.270 | 33.720 | 33.658~33.776 |
> | algebra, geometry, photosynthesis, cellular respiration, genetics | 36.288 | 36.133~36.484 | 33.741 | 33.622~33.826 |
>
> For HELM-MiCE, more generic words (e.g., subject) are clustered closer to the origin than more specific words (e.g., biology), which have a smaller norm than even more specific words (e.g., photosynthesis). However, this does not necessarily hold for the DeepseekV3 1B model, demonstrating how HELM-MiCE better handles semantic hierarchies. We will provide low-dimensional visualizations of these embeddings in our revision.
>
> Let us now consider tokens from questions from MMLU that HELM-MiCE correctly answers but DeepseekV3 gets wrong. We examine their final embedding norms:
>
> | **HELM-MiCE** | **HELM-MiCE** | **DeepseekV3** | **DeepseekV3** |
> | --- | --- | --- | --- |
> | Words | Norm Range | Words | Norm Range |
> | A, How, does, if, there, have, is, any, with, of | 36.031~36.396 | is, a, connecting, graph, there, edges, complete, have, of,  | 33.668~33.768 |
> | discrete, vertices, edges, connecting, pair, graph, complete, many , 10 | 36.506~36.717 | discrete, 10, how, if, pair, does, with, A, vertices, any | 33.772~33.908 |
>
> For HELM-MiCE, general words have a smaller norm (e.g., how, if) while the more specific “graph theory” keywords are further away from the origin (e.g., connecting, graph). However, for DeepseekV3, general and specific words are sometimes intertwined in the embedding space. The hierarchical organization of the space enables the HELM models to sometimes better navigate the embedding space and obtain better performance.
>
> **Q:** Are each expert’s curvature values fixed or learnable, and how sensitive are results to that choice?
>
> **R:** For our experiments with MiCE, the experts’ curvatures are trainable and initiated uniformly from -0.1 to -2.0 (e.g., with 4 experts, the curvature values were -0.1, -0.733, -1.367, -2.0). We did not observe the curvature to converge to values that were far from their initial values (within $\pm0.1$ of the initial values). This is presumably due to the wide range of initial curvatures, which we have selected to better capture the geometric variation from token to token. Our ablations in Table 2 demonstrate the effectiveness of variable curvatures for each expert.
>
> **Q:** All models are trained on only 5B tokens...do you expect the hyperbolic–Euclidean gap to widen, stay flat, or shrink when trained on ≥100 B tokens? And what stability issues could emerge at that scale?
>
> **R:**  The results for the 1B models from checkpoints taken at 4B and 5B tokens demonstrate that HELM-MiCE consistently outperforms the Euclidean baselines, as shown below:
>
> |  | Token Count | CommonsenseQA | HellaSwag | OpenbookQA | MMLU | ARC-Challenging | Avg |
> | --- | --- | --- | --- | --- | --- | --- | --- |
> | DeepseekV3 (1B) | 4B | 18.8 | 25.1 | 26.4 | 23.5 | 22.1 | 23.2 |
> | **HELM-MiCE (1B)** | **4B** | **19.5** | **26.0** | **27.0** | **25.6** | **22.9** | **24.3** |
> | DeepseekV3 (1B) | 4.5B | 19.0 | 26.2 | 27.2 | 23.6 | 22.6 | 23.7 |
> | **HELM-MiCE (1B)** | **4.5B** | **19.7** | **26.4** | **27.6** | **25.5** | **23.1** | **24.5** |
> | DeepseekV3 (1B) | 5B | 19.5 | 26.2 | 27.4 | 23.6 | 22.7 | 23.9 |
> | **HELM-MiCE (1B)** | **5B** | **19.8** | **26.5** | **28.4** | **25.9** | **23.7** | **24.9** |
>
> As we scale up data to 5B, the gap between HELM and Euclidean baselines did not show any behavior or signs of diminishing, and the outperformance remained consistent. While one can only fully answer the reviewer’s question by increasing the training corpus to 100B tokens, these results give us more confidence in HELM to outperform the Euclidean baselines at larger scales. Our work presents a stable foundation to achieve training hyperbolic LLMs at such scales, paving the way for future explorations. It is worth noting that Euclidean space will always incur significant distortion when embedding tree-like data regardless of dimensionality and token count [b], while hyperbolic space can achieve arbitrarily low distortion [a]. Thus, there is a theoretical upper bound on the performance of the Euclidean baselines at larger scales, which is not present for HELM models.
>
> In regard to stability, we have employed several techniques to ensure training stability and stable convergence of the losses. Notably, as described in section 5, our modified hyperbolic token embedding method – where the embeddings are learned along the space dimension – improves stability over prior methods.
>
> **Q:** What motivated using the Lorentz representation over alternatives like the Poincaré ball or Klein model, and did you observe concrete numerical or training-stability advantages?
>
> **R:** The Lorentz hyperboloid comes with a **wide availability of fully hyperbolic operations in the Lorentz space**. These are hyperbolic operations that are performed directly on the manifold without requiring repeated mapping to and from the tangent space, and are not generally available in the Poincaré ball or Klein model. Many prior works have found that these Lorentz operations have several advantages, including better computational efficiency, higher expressiveness, and fewer mapping errors [24, 48, 6, 4]. These prior works have found that the fully hyperbolic operations concretely improve numerical stability, whereas tangent space mappings can result in NaN/overflow [24, 4]. Our models are built on top of these efforts and operations (e.g., fully hyperbolic linear layer and residual connection).
>
> Our proposed methods also take advantage of properties that are specific to the Lorentz space, such as formulating of HoPE based on Lorentzian rotations and the interactions between the time and space dimensions. These properties enable us to prove theoretical guarantees for HoPE and hyperbolic RMSNorm that are comparable to the Euclidean counterparts.
>
> ***References***:
> [a] Sarkar, R. Low distortion delaunay embedding of trees in hyperbolic plane. In International Symposium on Graph Drawing, pp. 355–366. Springer, 2011.
>
> [b] Lee, J. R., Naor, A., and Peres, Y. Trees and markov convexity, 2007.
>
> [c] Matousek, J. Lectures on Discrete Geometry. Springer, 2002.

---

> > ### Comment · Area_Chair_Wts4 · 2025-08-04
> > **Review Deadline is Fast Approaching (July 2nd)**
> >
> > Dear Reviewer,
> >
> > The authors have replied to your concerns, and it would be very helpful if you could engage in a discussion to clarify your concerns.
> >
> > Best,
> >
> > AC

---

> > ### Comment · Area_Chair_Wts4 · 2025-08-05
> > **Reminder about Engaging in Discussion (Deadline Aug. 8th)**
> >
> > Dear Reviewer,
> >
> > This is a friendly reminder about engaging in the discussion.
> >
> > Best,
> >
> > AC

---

> > ### Comment · Reviewer_1cTB · 2025-08-05
> >
> > I'm satisfied with the response of the authors and raised my score. I would encourage the authors to make the scaling aspect of the model (wrt training tokens & model parameters) more clear in the revision, e.g. adding some plots.

---

> > > ### Author Response · Authors · 2025-08-06
> > >
> > > We appreciate the reviewer for acknowledging the strength of our rebuttal and raising the score. We will incorporate the discussion and results for model scaling in our revision as recommended by the reviewer.

---

### Note · Authors · 2025-08-11

We thank the reviewers for their valuable suggestions and the AC for facilitating the discussion.
Our work introduces **HELM**, the first fully hyperbolic LLM trained at billion-parameter and billion-token scale to leverage the scale-free distribution of tokens. HELM introduces architectural contributions **unanimously** appreciated by the reviewers: **Mixture-of-Curvature-Experts (MiCE)**, **Hyperbolic Multi-Head Latent Attention (HMLA)**, and **Hyperbolic Rotary Positional Encoding (HoPE)**. We provide extensive theoretical proofs and ablations for each proposed module, and benchmark results showing consistent gains over popular Euclidean LLM architectures (DeepSeekV3, LLaMA).

**The reviewers’ main concerns were:**

**(i).** Qualitative evidence showing why our proposed hyperbolic LLM outperform Euclidean baselines.

**(ii).** The scaling potential of HELM models in terms of parameter and token count.

**(iii).** Statistical significance and comparison to prior hyperbolic Transformers.

In our rebuttal, we addressed these concerns by presenting:

**(i).** Qualitative case studies demonstrating HELM’s superior semantic hierarchy representation and separation based on word specificity as compared to Euclidean baselines.

**(ii). For parameter scaling:** Computational and memory usage comparison between HELM and Euclidean baselines that confirms HELM’s potential as a scalable architecture; **For token count scaling:** Results across training stages showing consistent improvement over Euclidean baselines as token count increases.

**(iii).** **Comparing to prior hyperbolic Transformers:** Results that showed HELM consistently outperforming all of the previous hyperbolic Transformers; **Statistical significance:** Statistics that demonstrated consistent improvement over Euclidean baselines and statistical significance across multiple runs and training stages.

All rebuttal results and will be included in the camera-ready version. We thank the reviewers again for their constructive feedback.

---

### Decision · Program_Chairs · 2025-09-17

**Decision:**

Accept (poster)

**Comment:**

The submission presents HELM, a new class of Large Language Models designed to operate in hyperbolic space rather than traditional Euclidean geometry. By adapting key components like Mixture of Experts and Multi-head Latent Attention to this geometry, HELM aims to overcome limitations such as training instability and reduced generative performance. Experimental results show modest improvements over standard Euclidean-based models.

During the discussion with authors and reviewers, the issues that still remained unanswered are:

1. Scalability of the model (R-1cTB and R-Ar71): It is unclear how this model would scale with data and model parameters.
2. Marginal improvements (R-2Div): Experiments show marginal improvements over the baselines.
3. Comparisons of Hyperbolic and Euclidean norms (R-cdXH): Authors provided an unconvincing and/or unclear answer.

To solve some of the issues above required more time and resources (e.g., 1 and 2). While many issues remained open, the majority of reviewers agreed that the approach is novel and can be of interest to the community. Because of the submissions’ novelty, I am recommending its acceptance to the program.